# Rho-kinase inhibition improves haemodynamic responses and circulating ATP during hypoxia and moderate intensity handgrip exercise in healthy older adults

Matthew L. Racine[1] , Janée D. Terwoord[1], Nathaniel B. Ketelhut[1] , Nate P. Bachman[1], Jennifer C. Richards[1], Gary J. Luckasen[2] and Frank A. Dinenno[1]

[1]*Human Cardiovascular Physiology Laboratory, Department of Health and Exercise Science, Colorado State University, Fort Collins, CO, USA*
[2]*Medical Center of the Rockies, University of Colorado Health, Loveland, CO, USA*

Edited by: Michael Hogan & Bruno Grassi

Linked articles: This article is highlighted in a Perspective article by González-Alonso. To read this article, visit https://doi.org/10.1113/JP283322.

The peer review history is available in the Supporting information section of this article (https://doi.org/10.1113/JP282730#support-information-section).

**Matthew L. Racine** received his undergraduate degree from the University of Idaho (2009), MS from the University of Colorado Boulder (2012), and PhD from Colorado State University (2018). As a Pre-Doctoral Ruth L. Kirschstein National Research Service Award Fellow trained under Dr Frank Dinenno in the Human Cardiovascular Physiology Laboratory at CSU, his research on the mechanisms of skeletal muscle blood flow regulation during physiological stressors such as exercise and hypoxia in young and older adults focused on the contribution of red blood cells to this process through the release of ATP.

**Abstract** Skeletal muscle haemodynamics and circulating adenosine triphosphate (ATP) responses during hypoxia and exercise are blunted in older (OA) *vs.* young (YA) adults, which may be associated with impaired red blood cell (RBC) ATP release. Rho-kinase inhibition improves deoxygenation-induced ATP release from OA isolated RBCs. We tested the hypothesis that Rho-kinase inhibition (via fasudil) *in vivo* would improve local haemodynamic and ATP responses during hypoxia and exercise in OA. Healthy YA ($25 \pm 3$ years; $n = 12$) and OA ($65 \pm 5$ years; $n = 13$) participated in a randomized, double-blind, placebo-controlled, crossover study on two days ($\geq 5$ days between visits). A forearm deep venous catheter was used to administer saline/fasudil and sample venous plasma ATP ($[ATP]_V$). Forearm vascular conductance (FVC) and $[ATP]_V$ were measured at rest, during isocapnic hypoxia (80% $S_{pO_2}$), and during graded rhythmic handgrip exercise that was similar between groups (5, 15 and 25% maximum voluntary contraction (MVC)). Isolated RBC ATP release was measured during normoxia/hypoxia. With saline, $\Delta$FVC was lower ($P < 0.05$) in OA *vs.* YA during hypoxia ($\sim 60\%$) and during 15 and 25% MVC ($\sim 25$–30%), and these impairments were abolished with fasudil. Similarly, $[ATP]_V$ and ATP effluent responses from normoxia to hypoxia and rest to 25% MVC were lower in OA *vs.* YA and improved with fasudil ($P < 0.05$). Isolated RBC ATP release during hypoxia was impaired in OA *vs.* YA ($\sim 75\%$; $P < 0.05$), which tended to improve with fasudil in OA ($P = 0.082$). These data suggest Rho-kinase inhibition improves haemodynamic responses to hypoxia and moderate intensity exercise in OA, which may be due in part to improved circulating ATP.

(Received 12 December 2021; accepted after revision 9 May 2022; first published online 16 May 2022)

**Corresponding author** F. A. Dinenno: Department of Health and Exercise Science, Colorado State University, 220 Moby-B Complex, Fort Collins, CO 80523-1582, USA. Email: frank.dinenno@colostate.edu

**Abstract figure legend** Skeletal muscle haemodynamics and circulating ATP responses during hypoxia and exercise are blunted with age, which may be associated with impaired red blood cell (RBC) ATP release. Rho-kinase inhibition improves deoxygenation-induced ATP release from isolated RBCs of older adults. In a randomized, double-blind, placebo-controlled, crossover study, Rho-kinase inhibition via intravenous fasudil improved forearm haemodynamic and circulating ATP responses to systemic isocapnic hypoxia and moderate intensity rhythmic handgrip exercise in healthy older compared to young adults.

## Key points

- Skeletal muscle blood flow responses to hypoxia and exercise are impaired with age. Blunted increases in circulating ATP, a vasodilator, in older adults may contribute to age-related impairments in haemodynamics.
- Red blood cells (RBCs) are a primary source of circulating ATP, and treating isolated RBCs with a Rho-kinase inhibitor improves age-related impairments in deoxygenation-induced RBC ATP release.
- In this study, treating healthy older adults systemically with the Rho-kinase inhibitor fasudil improved blood flow and circulating ATP responses during hypoxia and moderate intensity handgrip exercise compared to young adults, and also tended to improve isolated RBC ATP release.
- Improved blood flow regulation with fasudil was also associated with increased skeletal muscle oxygen delivery during hypoxia and exercise in older adults.
- This is the first study to demonstrate that Rho-kinase inhibition can significantly improve age-related impairments in haemodynamic and circulating ATP responses to physiological stimuli, which may have therapeutic implications.

## Introduction

Cardiovascular disease (CVD) remains the leading cause of death worldwide and the majority of CVD-related mortality is associated with arterial dysfunction (Benjamin et al., 2017). Advancing age is the primary risk factor for CVD, and over 90% of all deaths associated with CVD are estimated to occur in adults over

60 years old (Benjamin et al., 2017). Furthermore, healthy (primary) ageing is associated with a decline in functional capacity that leads to reductions in exercise tolerance, functional independence and overall quality of life (WHO, 1993). All of these age-associated changes, as well as vascular pathologies like atherosclerosis and ischaemic disease, involve impairments in vascular control and the subsequent regulation of tissue blood flow and oxygen delivery.

The multifaceted nature of local blood flow regulation requires an integrated and coordinated balance between vasodilatory factors which can arise from the vascular endothelium, circulating elements in the blood, tissue metabolites, mechanical forces, and vasoconstricting signals from the sympathetic nervous system and vasculature (Clifford & Hellsten, 2004; Hellsten et al., 2012; Laughlin et al., 2012; Mortensen & Saltin, 2014; Segal, 2005). Primary ageing is associated with increases in sympathetic nervous system activity (reviewed by Dinenno & Joyner, 2006) and declines in the production or bioavailability of vasodilatory molecules, as well as reductions in skeletal muscle blood flow during physiological stimuli such as exercise (for review, see Hearon Jr. & Dinenno, 2016; Proctor & Parker, 2006; Wray & Richardson, 2015) and hypoxia (Casey et al., 2011; Richards et al., 2017). Of these alterations in vasoactive stimuli, attenuated local vasodilatory signalling is likely to be a major contributor to the age-related impairment in blood flow regulation, as data from our laboratory suggest that augmented sympathetic vasoconstriction does not contribute to the reduction in peripheral vasodilatation and skeletal muscle hyperaemia during hypoxia or handgrip exercise in older adults (Richards et al., 2014, 2017).

Among the vasodilatory signals affected by ageing, blunted increases in circulating adenosine triphosphate (ATP) during hypoxia and exercise (Kirby et al., 2012) may be one of the most significant impairments given the unique ability of circulating ATP to stimulate local and conducted vasodilatation via binding to purinergic $P_{2Y}$ receptors on the endothelium (Collins et al., 1998; Dora, 2017; Winter & Dora, 2007) while also blunting sympathetic $\alpha$-adrenergic vasoconstriction (Hearon Jr. et al., 2017; Kirby et al., 2008; Rosenmeier et al., 2004). Importantly, it has been demonstrated that the vasodilatory responsiveness to exogenous ATP is preserved in the forearm of older adults (Kirby et al., 2010), and although this may differ in other vascular beds (e.g. the leg) dependent on physical activity status (Mortensen et al., 2012), the collective evidence suggests that potential age-related impairments in the contribution of ATP to vascular control and regulation of skeletal muscle blood flow are related to the source of intravascular ATP.

In this context, red blood cells (RBCs) release ATP in response to cell deformation and in direct proportion to the degree of haemoglobin deoxygenation, and can therefore contribute to the coupling of blood flow and oxygen delivery to tissue metabolic demand (Bergfeld & Forrester, 1992; Dietrich et al., 2000; Ellsworth, 2000; Ellsworth & Sprague, 2012; Ellsworth et al., 1995; González-Alonso et al., 2002; Jagger et al., 2001; Jensen, 2009; Sprague et al., 2009). Furthermore, increases in circulating ATP during exercise are dependent on intact skeletal muscle perfusion (i.e. intravascular sources), further implicating the RBC as a primary source of intravascular ATP (Kirby et al., 2013). We recently demonstrated that increases in plasma ATP during hypoxia and exercise are impaired with advancing age, and this is likely due to impaired deoxygenation-induced ATP release from isolated RBCs of older adults (Kirby et al., 2012). We also recently demonstrated that reduced RBC deformability in older adults is a primary underlying mechanism of impaired deoxygenation-induced ATP release and that improving deformability via treatment of isolated RBCs from healthy older adults with a Rho-kinase inhibitor can restore their ability to release ATP in response to deoxygenation (Racine & Dinenno, 2019). However, whether Rho-kinase inhibition improves ATP release *in vivo* and haemodynamic responses to hypoxia and exercise in healthy older adults is unknown. Thus, the primary goal of the present study was to test the hypothesis that systemic administration of the Rho-kinase inhibitor fasudil would improve the haemodynamic responses to hypoxia and exercise in healthy older adults and that this would be accompanied by improvements in circulating ATP and deoxygenation-induced ATP release from isolated RBCs. We chose the forearm model to test our hypotheses given our prior findings of impaired forearm haemodynamic and plasma ATP responses during rhythmic handgrip exercise and systemic hypoxia in older adults (Kirby et al., 2012).

## Methods

### Ethical approval and subjects

With approval from the Institutional Review Board at Colorado State University (protocol 16-6361H) and after written informed consent, a total of 12 young and 13 older healthy adults participated in the present investigation. All subjects were free from overt cardiovascular disease as assessed from a medical history, free of cardiovascular medications, non-smokers, non-obese (body mass index <30 kg m$^{-2}$), normotensive (resting blood pressure <140/90), and sedentary to moderately active. Young female subjects were studied during the early follicular phase of their menstrual cycle to minimize any potential cardiovascular effects of sex-specific hormones, whereas older female subjects were post-menopausal and not on hormone replacement therapy. Additionally, older

subjects were further evaluated for clinical evidence of cardiopulmonary disease with a physical examination and resting and exercise (Balke protocol) electro-cardiograms. Body composition was determined by whole-body dual-energy X-ray absorptiometry scans (QDR series software, Hologic, Inc., Marlborough, MA, USA). Whole blood lipid panels were run using a Piccolo Xpress chemistry analyser (Abaxis, Union City, CA, USA). Studies were performed in the Human Cardiovascular Physiology Laboratory at Colorado State University (altitude: ∼1500 m) after an overnight fast, 24-h abstention from alcohol/substance use, and a 12-h abstention from caffeine and exercise, with subjects in the supine position with the experimental arm abducted to 90° and slightly elevated above heart level upon a tilt-adjustable table. All studies were performed according to the *Declaration of Helsinki*.

## Experimental design and general experimental protocol

The overall experimental design and timeline for each experimental visit is depicted in Fig. 1. Using a double-blind, placebo-controlled, crossover design, subjects were randomized to receive an infusion of either saline (placebo control) or the Rho-kinase inhibitor fasudil for their first experimental visit. Subjects

then received the opposite treatment for their second experimental visit, with at least 5 days and no more than 2 months between the first and second visit. Fasudil and hydroxyfasudil are metabolized quickly (half-life of ∼45 min and ∼280 min, respectively, with a 60 mg 60 min⁻¹ infusion of fasudil; Shibuya et al., 2005) and thus typically administered two to three times per day in clinical practice (Jiang et al., 2015; Satoh et al., 2014; Shibuya et al., 1992, 2005; Suzuki et al., 2007, 2008; Zhao et al., 2011). Therefore, at least 5 days between visits was deemed to be sufficient for washout of any potential effects of fasudil administration.

All experimental measures were performed in the same order for each visit within a subject, with the order of hypoxia and graded-intensity rhythmic handgrip exercise trials randomized and counterbalanced between subjects. For both the hypoxia and exercise trials, resting haemodynamics were measured for 2−3 min until a steady-state was observed, after which the physiological stimulus was initiated. The hypoxia trial consisted of 3 min of steady-state hypoxia at an oxygen saturation of ∼80% as assessed via pulse oximetry on the earlobe ($S_{pO_2}$; plus ∼2 min for the normoxia to hypoxia transition). The exercise trial consisted of 4 min at each workload to ensure that steady-state haemodynamics were achieved. Timing of blood sampling is indicated by arrows (Fig. 1*B*), with blood samples for plasma [ATP] taken under steady-state

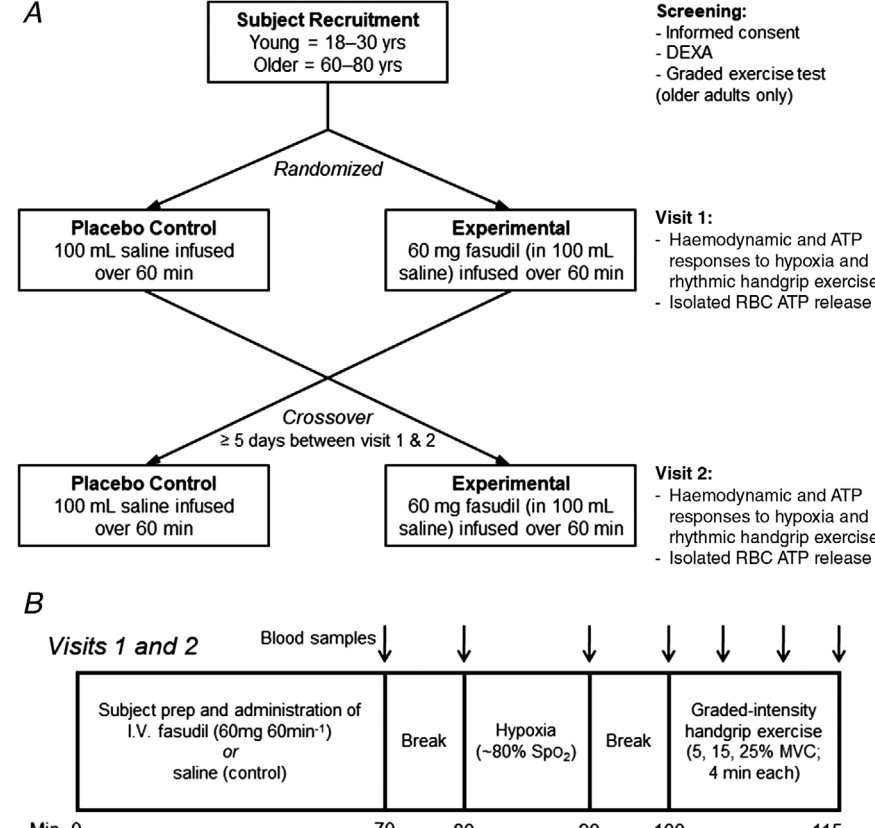

**Figure 1. Overall experimental design and experimental visit timeline**
*A*, double-blind, placebo-controlled, randomized, crossover experimental design. After screening, eligible subjects were randomized in a double-blind manner to receive an infusion of either saline (placebo control) or fasudil for their first experimental visit. Subjects then received the opposite treatment for their second experimental visit, with at least 5 days between visits. *B*, experimental visit timeline. After placement of the venous catheter and treatment infusion, experimental measures were made during systemic isocapnic hypoxia and graded-intensity rhythmic handgrip exercise trials; order of the trials was randomized and counterbalanced between subjects, but kept the same for both visits within a subject. Timing of blood sampling is indicated by arrows (Fig. 1B), with samples for plasma [ATP] taken under steady-state conditions at rest, during hypoxia and the end of each exercise workload.

conditions at rest, the end of hypoxia and the end of each exercise workload.

## Venous catheterization

As described previously by our laboratory (Crecelius et al., 2011, 2013; Kirby et al., 2012, 2013; Racine et al., 2018), an 18- or 20-gauge (depending on inspection of vein size) 5.1 cm catheter was inserted in retrograde fashion into an antecubital vein of the experimental arm for treatment administration and deep venous blood samples. The catheter was connected to a three-way stopcock, with one connection to an intravenous solution set for treatment administration followed by continuous flushing with saline at a rate of ~2 ml min$^{-1}$ for the duration of the study to keep it patent and the other connection to a 10- or 3-ml syringe for blood sampling. Sample size variability in some blood draw-dependent measures (e.g. ATP and blood gases) is due to loss of catheter patency in some participants.

## Intravenous fasudil and placebo (saline) administration

Fasudil monohydrochloride (fasudil; LC Laboratories, Woburn, MA, USA) was prepared in saline (10 mg ml$^{-1}$ sodium chloride 0.9% PF injection; Pencol Compounding Pharmacy, Denver, CO, USA) and passed all measures for purity by HPLC, sterility, endotoxins and fungal presence (Analytical Research Laboratories, Oklahoma City, OK, USA) prior to use. Sixty milligrams of fasudil (6 ml vial) was added to a 100 ml saline bag immediately prior to administration, covered to protect it from exposure to light, and infused intravenously over 60 min (Shibuya et al., 2005). This single dose of fasudil was well-tolerated by both young and older adults, and no adverse events were observed or reported in either age group. For the placebo control visit, administration of saline was identical to that of fasudil, with a covered 100 ml saline bag infused intravenously over 60 min. Fasudil mixed in saline was indistinguishable from saline alone, and thus all investigators and subjects remained blinded during treatment administration.

## Forearm blood flow and vascular conductance

A 12 MHz linear-array ultrasound probe (Vivid 7, GE Healthcare, Milwaukee, WI, USA) was used to determine brachial artery mean blood velocity (MBV) and diameter proximal to the catheter insertion site as described previously (Crecelius et al., 2011; Hearon Jr. et al., 2017; Kirby et al., 2012; Racine et al., 2018; Richards et al., 2017). Foam tape was used to mark the outline of the probe for consistent placement and measurement over the course of the experiments. For blood velocity measurements,

the probe insonation angle was maintained at <60° and the frequency used was 5 MHz. The Doppler shift frequency spectrum was analysed via a Multigon 500 M TCD (Multigon Industries, Mt Vernon, NY, USA) spectral analyser from which MBV was determined as a weighted mean of the spectrum of Doppler shift frequencies. Brachial artery diameter measurements were made in duplex mode at end-diastole and between contractions (at least in triplicate) during steady-state conditions (Crecelius et al., 2011; Kirby et al., 2012; Racine et al., 2018; Richards et al., 2017). Forearm blood flow (FBF) was calculated as: FBF = MBV × $\pi$ × (brachial artery diameter/2)$^2$ × 60, where FBF is expressed in ml min$^{-1}$, MBV in cm s$^{-1}$, brachial diameter in cm, and 60 was used to convert from ml s$^{-1}$ to ml min$^{-1}$. Forearm vascular conductance (FVC) was calculated as (FBF MAP$^{-1}$) × 100 and expressed in ml min$^{-1}$ 100 mmHg$^{-1}$ (Hearon Jr. et al., 2017; Kirby et al., 2012; Racine et al., 2018; Richards et al., 2014, 2017). Haemodynamic data are presented as absolute values as well as the response from baseline ($\Delta$) to take into account individual differences in baseline FBF and FVC. All studies were performed in a semi-darkened, cool (20–22°C), temperature-controlled environment with a fan directed toward the forearm to minimize the contribution of skin blood flow to forearm haemodynamics.

## Systemic isocapnic hypoxia

The systemic isocapnic hypoxia trial was performed using a self-regulating partial rebreathe system developed by Banzett et al. (2000) and more recently described by our laboratory (Crecelius et al., 2011; Kirby et al., 2012; Markwald et al., 2011; Racine et al., 2018; Richards et al., 2017). This system allows for constant alveolar fresh air ventilation independent of changes in breathing frequency or tidal volume (Banzett et al., 2000; Dinenno et al., 2003; Wilkins et al., 2008). Using this system, we were able to clamp end-tidal $CO_2$ levels despite the hypoxia-induced increases in ventilation. The level of oxygen was manipulated by mixing nitrogen with medical air via an anaesthesia gas blender. Specifically, inspired oxygen was titrated to achieve an $S_{pO_2}$ of ~80%. Subjects breathed through a scuba mouthpiece with a nose-clip to prevent nasal breathing. An anaesthesia monitor (Cardiocap/5, Datex-Ohmeda, Louisville, CO, USA) was used to determine heart rate (HR; 3-lead ECG) and expired $CO_2$ sampled at the mouthpiece. Ventilation was measured via a turbine pneumotachograph (model 17125 UVM, Vacu-Med, Ventura, CA, USA).

## Graded-intensity rhythmic handgrip exercise

Maximum voluntary contraction (MVC) was determined for the experimental arm as the average of three maximal

squeezes of a handgrip dynamometer (Stoelting, Chicago, IL, USA) that were within 3% of each other. Rhythmic handgrip exercise during the trials was performed with weights corresponding to 5%, 15% and 25% MVC ($\sim$15%, 40% and 70% of maximal work rate, respectively; Richards et al., 2014) attached to a pulley system and lifted 4−5 cm over the pulley at a duty cycle of 1 s contraction−2 s relaxation (20 contractions per min) using both visual and auditory feedback to ensure the correct timing (Dinenno & Joyner, 2003, 2004; Kirby et al., 2012; Richards et al., 2014). Handgrip exercise was performed for 4 min at each workload, for a total of 12 min.

### Blood sampling and measurement of [fasudil], [hydroxyfasudil], plasma [ATP], plasma [Hb] and blood gases

Timing of deep venous blood samples is indicated by arrows in Fig. 1. Based on preliminary pharmacokinetics experiments performed in our laboratory (data not shown), a blood sample for peak plasma concentrations of fasudil and hydroxyfasudil was taken $\sim$15 min after the treatment infusion ended and an additional sample was taken at the end of the study immediately prior to catheter removal to confirm that concentrations of each compound remained at a level that can effectively inhibit Rho-kinase, which has been shown to range from 0.08 to 1.9 $\mu$M for fasudil (Davies et al., 2000; Jacobs et al., 2006; Rikitake et al., 2005; Satoh et al., 2012; Shibuya et al., 2005; Wickman et al., 2003) and from 0.04 to 1.8 $\mu$M for hydroxyfasudil (Jacobs et al., 2006; Rikitake et al., 2005; Satoh et al., 2012; Shibuya et al., 2005; Shimokawa, 2002; Shimokawa et al., 1999).

The plasma concentrations of fasudil and hydroxyfasudil were measured by the Colorado State University Analytical Resources Core using triple quadrupole UPLC-MS/MS as described by Chen et al. (2010). Briefly, stock solutions of standard fasudil (1 mg ml$^{-1}$), hydroxyfasudil (0.5 mg ml$^{-1}$) and ranitidine (1 mg ml$^{-1}$) were prepared in methanol. A nine-point calibration curve was prepared with fasudil and hydroxyfasudil using ranitidine as an internal standard in methanol, and in a pooled blank plasma created by mixing aliquots of plasma from test subjects following saline injection. Analytical samples and matrix calibration solutions were prepared by mixing a 230 $\mu$l aliquot of plasma, 400 $\mu$l methanol and 70 $\mu$l of a 5 $\mu$g ml$^{-1}$ solution of ranitidine in an Eppendorf tube; they were vortexed for 30 s, and centrifuged at 11,000 rpm for 5 min; 150 $\mu$l of the supernatant was transferred to an autosampler vial. Quality control samples and blanks were run every eight injections and all injections were introduced in duplicate. Data were processed using the response ratio for target analytes fasudil and hydroxyfasudil to the internal standard ranitidine.

Blood samples for plasma [ATP] and blood gases were taken immediately after the treatment infusion (ATP standard curve sample) and at the end of rest, hypoxia and each exercise intensity. Our method for blood sampling, preparation and measurement of plasma [ATP] (Crecelius et al., 2013; Kirby et al., 2012, 2013) generally follows the procedures established by Gorman and colleagues (Gorman et al., 2003, 2007) and was performed as previously described in detail (Kirby et al., 2012). Briefly, $\sim$3–5 ml of venous blood was drawn directly into a pre-heparinized 10 ml syringe, from which 2 ml was gently and at once expelled into a tube containing 2.7 ml of an ATP-stabilizing solution to equal a blood:diluent ratio of 1.35 (Crecelius et al., 2013; Gorman et al., 2003, 2007; Kirby et al., 2012, 2013). This ATP-stabilizing solution is used to inhibit the degradation of ATP via nucleotidases and additional ATP release post-sampling. The blood-diluent mixture was immediately centrifuged at $\sim$1200 $g$ (4000 rpm) for 3 min at 22°C, and 100 $\mu$l of the supernatant was taken for measurement of plasma [ATP] via luciferin-luciferase assay. An ATP standard curve was created on each visit prior to hypoxia and exercise trials using plasma from each subject studied as the medium. All plasma ATP measures were performed at least in triplicate.

To account for the potential influence of RBC haemolysis on measures of plasma [ATP], which can increase significantly with only small amounts of haemolysis, a 1 ml sample of plasma supernatant from the blood–diluent mixture was taken immediately following ATP measurements and analysed for plasma haemoglobin (Hb) via spectrophotometry (SpectraMax, Molecular Devices, Sunnyvale, CA, USA) at wavelengths of 415, 380 and 450 nm as described previously by our laboratory (Crecelius et al., 2013; Kirby et al., 2012, 2013). The percentage haemolysis was then calculated as [(100 − haematocrit) $\times$ plasma [Hb] total [Hb]$^{-1}$] $\times$ 100. Any sample that was more than 2 standard deviations from the mean percentage haemolysis was excluded from the analysis and regarded as a technical error.

Blood gas samples ($\sim$2 ml) were immediately ($<$1 min) analysed with a clinical blood gas analyser (Rapid Point 400 Series Automatic Blood Gas System, Siemens Healthcare Diagnostics, Deerfield, IL, USA) for partial pressures of oxygen and carbon dioxide ($P_{O_2}$ and $P_{CO_2}$), pH, fraction of oxygenated haemoglobin ($F_{O_2 Hb}$), oxygen content and Hb.

### RBC isolation and deoxygenation

Blood was collected from the antecubital vein catheter into two Vacutainer tubes containing sodium heparin (158 USP units). Whole blood was centrifuged (500 $g$, 4°C, 10 min) followed by removal of the plasma and buffy coat by aspiration. All RBC ATP measures were performed

immediately after RBC isolation was completed. RBCs were diluted to 20% haematocrit with a bicarbonate-based buffer containing (in mM): 4.7 KCl, 2.0 CaCl$_2$, 1.2 MgSO$_4$, 140.5 NaCl, 11.1 glucose, 23.8 NaHCO$_3$, and 0.5% BSA warmed to 37°C. As described previously by our laboratory (Kirby et al., 2012; Racine & Dinenno, 2019), these 20% haematocrit RBC suspensions were placed in a rotating bulb tonometer (Eschweiler GmbH & Co. KG, Germany) and incubated for 30 min in normoxia (16% O$_2$, 6% CO$_2$, balance nitrogen; $P_{O_2}$ ~123 mmHg and $F_{O_2 Hb}$ ~95% across all age groups and conditions) at 37°C. RBC samples were removed from the tonometer bulb for measurement of extracellular and intracellular ATP in normoxia (details below). RBCs were then deoxygenated by exposure to hypoxia (1.5% O$_2$, 6% CO$_2$, balanced nitrogen; $P_{O_2}$ ~25 mmHg and $F_{O_2 Hb}$ ~34% across all age groups and conditions) for 15 min, the goal of which was to reduce $P_{O_2}$ and $F_{O_2 Hb}$ to levels within the range observed *in vivo* during conditions such as exercise (Kirby et al., 2012), and RBC samples were taken for measurement of ATP as in normoxia. Normoxic and hypoxic gases were blended with a gas blender (MCQ Gas Blender Series 100, Rome, Italy) and humidified before introduction into the tonometer bulbs. Blood gases were confirmed by blood gas analysis (Siemens Rapid Point 405 Series Automatic Blood Gas System, Los Angeles, CA, USA) (Kirby et al., 2012; Racine & Dinenno, 2019).

## Measurements of extracellular ATP and RBC total intracellular ATP

ATP was measured by the luciferin–luciferase technique as described previously (Kirby et al., 2012; Racine & Dinenno, 2019; Richards et al., 2013; Sprague et al., 2001; Sridharan, Adderley et al., 2010; Sridharan, Sprague et al., 2010; Thuet et al., 2011), with light emission during the reaction detected by a luminometer (TD 20/20, Turner Designs, Sunnyvale, CA, USA). For measurement of extracellular ATP (i.e. ATP release), a 10 $\mu$l sample of the 20% haematocrit suspension was taken from the tonometer bulb and diluted 500-fold (0.04% haematocrit), from which a 200 $\mu$l sample was taken and injected into a cuvette containing 100 $\mu$l of firefly tail extract (10 mg ml$^{-1}$ deionized water; Sigma-Aldrich, St Louis, MO, USA) and 100 $\mu$l of D-luciferin (0.5 mg ml$^{-1}$ deionized water; Research Products International, Mount Prospect, IL, USA). Peak light output was measured at least in triplicate for each experimental condition and the mean was used for determination of ATP levels by comparison to a standard curve for ATP (Calbiochem, San Diego, CA, USA) generated on the day of the experiment. Cell counts were obtained from each 0.04% RBC suspension and extracellular ATP was normalized to 4 × 10$^8$ cells. To confirm that ATP release was not due to haemolysis,

the 0.04% RBC suspensions from which samples for ATP analysis and cell counting were taken were analysed for free haemoglobin by measuring absorbance at 405 nm similar to previous reports (Kirby et al., 2012, 2014; Racine & Dinenno, 2019; Richards et al., 2013; Sprague et al., 1998, 2011; Sridharan, Adderley et al., 2010; Thuet et al., 2011) as well as at 570 nm and subtracting out the background at 700 nm as recently suggested by Keller and colleagues (Keller et al., 2017), and samples with significant lysis were excluded, which occurred in less than 1% of samples (Racine & Dinenno, 2019).

To determine if donor age or *in vivo* fasudil administration affected total intracellular ATP or the increase in RBC glycolytic activity during hypoxia (Campanella et al., 2005; Kirby et al., 2014; Lewis et al., 2009; Messana et al., 1996; Racine & Dinenno, 2019), 50 $\mu$l samples of the 20% haematocrit RBC suspension were taken from the tonometer bulb in normoxia and hypoxia following measurement of extracellular ATP and lysed in deionized water at room temperature (a 20-fold dilution). This lysate was diluted an additional 400-fold (8000-fold total) and ATP was measured by the same approach used for determination of extracellular ATP (Kirby et al., 2012, 2014; Racine & Dinenno, 2019; Sprague et al., 2011; Sridharan, Adderley et al., 2010; Sridharan, Sprague et al., 2010; Thuet et al., 2011). Values were normalized to ATP concentration per RBC.

## Data acquisition and analysis

All *in vivo* data were collected and stored on a computer at 250 Hz and were analysed offline with signal-processing software (WinDaq, DATAQ Instruments, Akron, OH, USA). Resting MAP was determined non-invasively over the brachial artery (Cardiocap/5). Beat-by-beat MAP was measured at the heart level by finger photoplethysmography (Finometer, FMS, Amsterdam, Netherlands) on the middle finger of the control hand during hypoxia and rhythmic handgrip exercise trials (Kirby et al., 2012). FBF, HR, MAP, and oxygen saturations (pulse oximetry) represent an average of the last 30 s of the appropriate time period. Minute ventilation and end-tidal CO$_2$ in the hypoxia trial were determined from an average of the data over the last minute of each time period in order to ensure an adequate number of sampling points.

Oxygen delivery, extraction, and consumption were quantified using venous oxygen content ($C_{vO_2}$) determined from deep venous blood samples taken at the end of rest, hypoxia, and each exercise intensity along with estimates of arterial oxygen content ($C_{aO_2}$) calculated as previously published by our laboratory (Richards et al., 2018): $C_{aO_2} = (\text{Hb} \times 1.36 \times S_{aO_2}) + P_{aO_2} \times 0.003$, where $S_{aO_2}$ was measured via pulse oximetry and $P_{aO_2}$ during normoxia (at rest in hypoxia trial and throughout the

duration of the exercise trial) and hypoxia (80% $S_{aO_2}$) was estimated based on previously published arterial blood gas data in young and older adults in our laboratory (Richards et al., 2017). Importantly, several studies including those from our laboratory have shown that arterial oxygen content does not change during forearm (handgrip) exercise in humans (Casey et al., 2010; Crecelius et al., 2011). Arteriovenous oxygen difference was calculated as $C_{aO_2} - C_{vO_2}$. Oxygen delivery was calculated as $C_{aO_2} \times$ FBF $\times$ 0.001 and expressed in ml min$^{-1}$. Oxygen extraction, reported as a percentage, was calculated as $(C_{aO_2} - C_{vO_2})$ $C_{aO_2}^{-1} \times 100$. Oxygen consumption across the forearm $(\dot{V}_{O_2})$ was calculated as $(C_{aO_2} - C_{vO_2}) \times$ FBF $\times$ 0.001 and expressed in ml min$^{-1}$.

To account for changes in FBF and its impact on [ATP] concentration measurements and to quantify the rate of total ATP draining the active muscle, ATP effluent was calculated as FBF $\times$ [ATP] $\times$ 0.001, as quantified previously by our laboratory (Crecelius et al., 2013; Kirby et al., 2012) and similar to other methods of data quantification when blood flow is altered (Giannarelli et al., 2009; González-Alonso et al., 2002).

### Statistics

All values are reported as means $\pm$ SD. All analyses were performed using R (R Core Team 2016, R Foundation for Statistical Computing, Vienna, Austria) with the lme4, lmerTest, pbkrtest and lsmeans packages. Age (young or older), drug (saline or fasudil), condition (rest or exercise intensity or hypoxia), and age $\times$ drug $\times$ condition for three-way repeated measures or age $\times$ drug for two-way repeated measures ANOVA were treated as fixed effects. In order to account for the crossover design, subject and subject $\times$ drug were included in the model as random effects for the three-way repeated measures analyses and subject was included as a random effect for the two-way repeated measures analyses. When an interaction or main effect was found, appropriate pairwise comparisons were made and a Tukey test was included when necessary. Comparisons of variables relative to zero were tested using a one-tailed Student's *t*-test, and differences in subject characteristics were tested using a two-tailed *t*-test. Comparisons between saline and fasudil were performed within age group and comparisons between young and old were performed within condition. Significance was set at $P < 0.05$.

### Results

#### Subject characteristics and plasma [fasudil] and [hydroxyfasudil]

Subject characteristics are reported in Table 1. The mean age difference between the young and older adults was

**Table 1. Subject characteristics**

| | Young | Older |
|---|---|---|
| Male:female | 6:6 | 6:7 |
| Age (years) | 25 ± 3 | 65 ± 5* |
| Body mass index (kg m$^{-2}$) | 23.1 ± 2.2 | 25.2 ± 3.5 |
| Body fat (%) | 24.5 ± 7.2 | 33.6 ± 8.5* |
| Forearm volume (ml) | 879.6 ± 255.3 | 1002.1 ± 339.2 |
| Forearm fat-free mass (g) | 733.7 ± 287.3 | 726.7 ± 293.2 |
| Maximum voluntary contraction (kg) | 33 ± 11 | 29 ± 10 |
| 5% workload (kg) | 1.6 ± 0.5 | 1.5 ± 0.5 |
| 15% workload (kg) | 4.9 ± 1.7 | 4.4 ± 1.5 |
| 25% workload (kg) | 8.1 ± 2.7 | 7.4 ± 2.5 |
| Total cholesterol (mg dl$^{-1}$) | 153 ± 31 | 182 ± 41 |
| LDL cholesterol (mg dl$^{-1}$) | 83 ± 22 | 106 ± 32 |
| HDL cholesterol (mg dl$^{-1}$) | 53 ± 12 | 59 ± 13 |
| LDL:HDL | 1.6 ± 0.4 | 1.8 ± 0.6 |
| Triglycerides (mg dl$^{-1}$) | 83 ± 28 | 89 ± 29 |

*$P < 0.05$ *vs.* young (within condition).

40 years. Older adults had a significantly higher body fat percentage compared to young adults ($P = 0.009$), but all other characteristics were similar, including forearm fat-free mass ($P = 0.952$), MVC ($P = 0.461$) and all exercise workloads ($P = 0.463–0.563$). In a subset of subjects ($n = 14$; seven young and seven older adults), blood samples were taken approximately 15 min after the treatment infusion stopped (peak) and at the end of the study just before the venous catheter was removed (end) for measurement of plasma concentrations of fasudil and hydroxyfasudil. The average [fasudil] at peak and end was $1.003 \pm 1.199$ and $0.071 \pm 0.038$ $\mu$M, respectively, and the average [hydroxyfasudil] at peak and end was $2.208 \pm 0.709$ and $1.013 \pm 0.499$ $\mu$M, respectively. Importantly, there were no differences between young and older adults, and fasudil was well-tolerated in all subjects with no adverse events.

#### Effects of age and fasudil on ventilation, haemodynamics and plasma ATP during systemic isocapnic hypoxia

Haemodynamics and ventilatory parameters during normoxia and systemic isocapnic hypoxia are reported in Table 2. $S_{pO_2}$, minute ventilation and end-tidal $CO_2$ were not significantly different between age groups in the saline condition, and while there were small age-group differences in $S_{pO_2}$ and minute ventilation in the fasudil condition, ~80% $S_{pO_2}$ was achieved in

**Table 2. Haemodynamics and ventilatory parameters during systemic hypoxia**

| | Young | | Older | |
|---|---|---|---|---|
| | Normoxia | Hypoxia | Normoxia | Hypoxia |
| **Saline** | | | | |
| MAP (mmHg) | $88 \pm 7$ | $89 \pm 6$ | $99 \pm 10^{\dagger}$ | $98 \pm 14^{\dagger}$ |
| HR (beats $min^{-1}$) | $57 \pm 7$ | $78 \pm 12$ | $59 \pm 7$ | $68 \pm 8^{\dagger}$ |
| $P_{vO_2}$ (mmHg) | $28.1 \pm 5.3$ | $26.6 \pm 2.1$ | $26.7 \pm 4.8$ | $25.8 \pm 4.5$ |
| $P_{vCO_2}$ (mmHg) | $43.8 \pm 3.1$ | $41.8 \pm 2.4$ | $45.2 \pm 3.3$ | $43.4 \pm 3.3$ |
| a-v$O_2$ (ml $l^{-1}$) | $106.5 \pm 26.3$ | $74.4 \pm 12.8^{\ddagger}$ | $103.7 \pm 23.0$ | $75.6 \pm 22.1^{\ddagger}$ |
| $[Hb]_V$ (g $dl^{-1}$) | $15.0 \pm 1.0$ | $15.1 \pm 0.9$ | $14.3 \pm 1.0$ | $14.1 \pm 1.0$ |
| $S_{pO_2}$ (%) | $98.6 \pm 1.1$ | $79.1 \pm 3.3$ | $97.7 \pm 1.5$ | $79.7 \pm 2.2$ |
| Minute ventilation (l $min^{-1}$; BTPS) | $8.5 \pm 2.2$ | $16.6 \pm 3.3$ | $8.5 \pm 3.4$ | $14.4 \pm 3.1$ |
| End-tidal $CO_2$ (mmHg) | $40.7 \pm 2.8$ | $39.9 \pm 2.9$ | $38.3 \pm 4.0$ | $37.6 \pm 3.0$ |
| **Fasudil** | | | | |
| MAP (mmHg) | $87 \pm 8$ | $90 \pm 12$ | $92 \pm 9^{*}$ | $92 \pm 10^{*}$ |
| HR (beats $min^{-1}$) | $59 \pm 10$ | $79 \pm 11$ | $57 \pm 6$ | $67 \pm 10^{\dagger}$ |
| $P_{vO_2}$ (mmHg) | $31.6 \pm 6.6$ | $25.1 \pm 2.3$ | $28.8 \pm 3.0$ | $26.3 \pm 4.8$ |
| $P_{vCO_2}$ (mmHg) | $43.0 \pm 5.5$ | $43.2 \pm 4.2$ | $46.1 \pm 3.8$ | $42.5 \pm 3.8^{\ddagger}$ |
| a-v$O_2$ (ml $l^{-1}$) | $92.4 \pm 34.7$ | $70.2 \pm 16.5$ | $91.3 \pm 12.9$ | $66.7 \pm 20.8^{\ddagger}$ |
| $[Hb]_V$ (g $dl^{-1}$) | $14.8 \pm 1.4$ | $14.7 \pm 1.3$ | $14.0 \pm 0.8$ | $13.6 \pm 0.9$ |
| $S_{pO_2}$ (%) | $98.7 \pm 1.5$ | $77.5 \pm 3.4$ | $97.3 \pm 2.0$ | $79.6 \pm 2.5^{\dagger}$ |
| Minute ventilation (l $min^{-1}$; BTPS) | $8.9 \pm 2.8$ | $19.0 \pm 5.0^{*}$ | $9.0 \pm 4.0$ | $13.7 \pm 3.7^{\dagger}$ |
| End-tidal $CO_2$ (mmHg) | $39.6 \pm 3.5$ | $38.8 \pm 3.1$ | $37.3 \pm 5.7$ | $37.2 \pm 3.9$ |

$n = 11$ for young (6M/5F).

$^{*}P < 0.05$ *vs.* saline.

$^{\dagger}P < 0.05$ *vs.* young (within condition).

$^{\ddagger}P < 0.05$ *vs.* normoxia (blood gases only).

Abbreviations: a-v$O_2$, arteriovenous oxygen difference; BTPS, body temperature and pressure (saturated); $[Hb]_V$, venous haemoglobin concentration; HR, heart rate; MAP, mean arterial pressure; $P_{vCO_2}$, venous partial pressure of carbon dioxide; $P_{vO_2}$, venous partial pressure of oxygen; $S_{pO_2}$, peripheral oxygen saturation.

both conditions and age groups. MAP was significantly higher in older *vs.* young adults during normoxia and hypoxia with saline. Fasudil significantly reduced MAP in older adults and abolished the age-group differences during normoxia and hypoxia ($P = 0.004$ and 0.008; Table 2). There were no differences in baseline (normoxia) FBF or FVC between age groups or treatment conditions (Fig. 2*A* and *B*). The increase in FBF and FVC from normoxia to hypoxia was lower in older *vs.* young adults in the saline control condition ($\Delta$FBF $2.1 \pm 2.1$ *vs.* $6.3 \pm 3.9$ ml $min^{-1}$ and $\Delta$FVC $2.8 \pm 3.3$ *vs.* $7.1 \pm 4.6$ ml $min^{-1}$ 100 $mmHg^{-1}$, respectively; $P = 0.031$ and 0.028). Fasudil significantly improved FBF and FVC responses to hypoxia in older adults only, such that there was no longer a difference between older and young adults ($\Delta$FBF $8.4 \pm 6.9$ *vs.* $6.2 \pm 4.0$ ml $min^{-1}$ and $\Delta$FVC $9.0 \pm 6.3$ *vs.* $6.2 \pm 3.6$ ml $min^{-1}$ 100 $mmHg^{-1}$, respectively; $P = 0.250$ and 0.147) (Fig. 2*C* and *D*).

Resting venous plasma [ATP] and ATP effluent in normoxia were not different between older and young adults in saline ($90.3 \pm 54.4$ *vs.* $79.9 \pm 48.3$ nmol $l^{-1}$ and $1.9 \pm 1.2$ *vs.* $1.6 \pm 1.2$ nmol $min^{-1}$, respectively; $P = 0.679$ and 0.604) or fasudil ($71.5 \pm 37.4$ *vs.* $81.3 \pm 50.3$ nmol $l^{-1}$ and $1.7 \pm 1.0$ *vs.* $1.5 \pm 0.8$ nmol $min^{-1}$, respectively; $P = 0.695$ and 0.763) conditions, and within each age group these values were not different between saline and fasudil ($P = 0.346$–$0.951$) (Fig. 2*E* and *F*). In older adults, $[ATP]_V$ and ATP effluent during hypoxia did not significantly increase *vs.* normoxia with saline ($95.7 \pm 67.8$ *vs.* $90.3 \pm 54.4$ nmol $l^{-1}$ and $2.2 \pm 1.4$ *vs.* $1.9 \pm 1.2$ nmol $min^{-1}$, respectively; $P = 0.574$ and 0.251), whereas with fasudil the increase during hypoxia *vs.* normoxia tended to be higher for $[ATP]_V$ ($88.8 \pm 74.1$ *vs.* $71.5 \pm 37.4$ nmol $l^{-1}$, respectively; $P = 0.073$) and was significantly higher for ATP effluent ($2.9 \pm 2.1$ *vs.* $1.7 \pm 1.0$ nmol $min^{-1}$, respectively; $P < 0.0001$) (Fig. 2*E* and *F*). In young adults, $[ATP]_V$ increased from normoxia to hypoxia with saline but not fasudil, whereas ATP effluent increased in both conditions (Fig. 2*E* and *F*). Sample size variability between haemodynamics and plasma ATP measures is due to loss of catheter patency in some participants.

### Effects of age and fasudil on haemodynamics and plasma ATP during graded-intensity rhythmic handgrip exercise

Haemodynamics and blood gases at baseline and during graded-intensity rhythmic handgrip exercise are reported in Table 3. MAP was significantly higher in older *vs.* young adults at all time points with saline ($P = 0.001$–$0.034$), but was significantly reduced with fasudil in the older adults such that there was no longer an age-related difference (Table 3; $P = 0.005$–$0.038$). Resting FBF and FVC were unaffected by age or fasudil (Fig. 3*A* and *B*; $P = 0.807$–$0.908$ and $0.743$–$0.977$). With saline, the FVC response from rest to exercise was lower in older *vs.* young

adults at 15% and 25% MVC ($\Delta$FVC $142.0 \pm 52.2$ *vs.* $187.8 \pm 59.0$ ml min$^{-1}$ 100 mmHg$^{-1}$ and $228.8 \pm 71.3$ *vs.* $329.9 \pm 87.1$ ml min$^{-1}$ 100 mmHg$^{-1}$, respectively; $P = 0.045$ and $<0.0001$; Fig. 3*B* and *D*), as was the FBF response at 25% MVC ($\Delta$FBF $255.3 \pm 71.6$ *vs.* $315.4 \pm 107.3$ ml min$^{-1}$; $P = 0.023$; Fig. 3*A* and *C*). When absolute haemodynamics are normalized to workload to account for possible individual differences between young and older participants, no significant differences were observed in FBF at any exercise intensity whereas FVC was significantly lower in older adults at 25% MVC ($35.9 \pm 12.2$ *vs.* $45.7 \pm 9.6$ ml min$^{-1}$ 100 mmHg$^{-1}$ kg$^{-1}$, respectively; $P = 0.038$). Fasudil significantly improved the FBF and FVC responses to

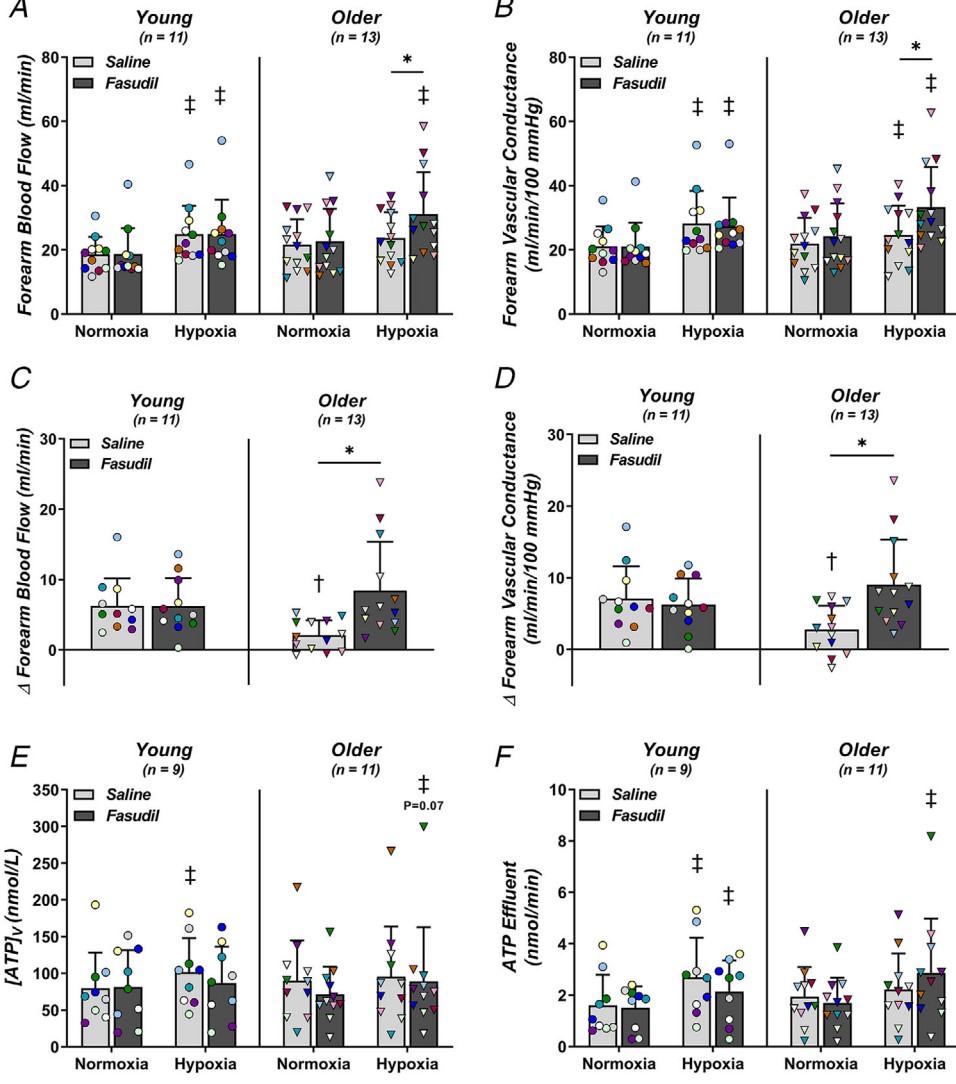

**Figure 2. Effects of age and fasudil on forearm haemodynamics and plasma ATP during systemic hypoxia**
*A* and *B*, fasudil significantly increased absolute forearm blood flow (FBF) and vascular conductance (FVC) in older adults during hypoxia; differences between age groups were not significant. *C* and *D*, age-related impairments in $\Delta$FBF and $\Delta$FVC responses from normoxia to hypoxia were reversed with fasudil. *E* and *F*, venous plasma ATP ([ATP]$_V$) and ATP effluent increased *vs.* normoxia in older adults with fasudil, but not saline; differences between age groups were not significant. *$P < 0.05$ *vs.* saline; †$P < 0.05$ *vs.* young; ‡$P < 0.05$ *vs.* normoxia.

**Table 3. Haemodynamics and blood gases during graded-intensity rhythmic handgrip exercise**

| | Young | | | | Older | | | |
|---|---|---|---|---|---|---|---|---|
| | Rest | 5% MVC | 15% MVC | 25% MVC | Rest | 5% MVC | 15% MVC | 25% MVC |
| **Saline** | | | | | | | | |
| MAP (mmHg) | $87 \pm 9$ | $90 \pm 12$ | $90 \pm 14$ | $94 \pm 18$ | $100 \pm 11^{\dagger}$ | $102 \pm 15^{\dagger}$ | $106 \pm 17^{\dagger}$ | $112 \pm 18^{\dagger}$ |
| HR (beats min$^{-1}$) | $57 \pm 9$ | $61 \pm 9$ | $63 \pm 9$ | $67 \pm 8$ | $58 \pm 8$ | $60 \pm 7$ | $62 \pm 7$ | $63 \pm 7$ |
| $P_{vO_2}$ (mmHg) | $25.7 \pm 6.2$ | $20.3 \pm 2.1$ | $24.5 \pm 4.1$ | $26.3 \pm 4.4$ | $27.3 \pm 8.1$ | $18.8 \pm 3.0$ | $22.4 \pm 3.5$ | $25.0 \pm 3.5$ |
| $P_{vCO_2}$ (mmHg) | $43.3 \pm 2.7$ | $49.5 \pm 3.5$ | $51.9 \pm 5.4$ | $54.9 \pm 7.1$ | $43.5 \pm 5.5$ | $48.8 \pm 4.8$ | $53.9 \pm 6.8$ | $54.4 \pm 6.8$ |
| a-vO$_2$ (ml l$^{-1}$) | $117.1 \pm 27.2$ | $142.2 \pm 11.1$ | $132.5 \pm 15.2$ | $126.0 \pm 19.0$ | $107.0 \pm 30.5$ | $139.7 \pm 17.4$ | $132.3 \pm 20.7$ | $121.5 \pm 21.0$ |
| [Hb]$_V$ (g dl$^{-1}$) | $14.8 \pm 1.0$ | $14.6 \pm 1.0$ | $14.9 \pm 1.1$ | $15.0 \pm 1.1$ | $14.4 \pm 1.1$ | $14.2 \pm 1.1$ | $14.4 \pm 1.1$ | $14.4 \pm 0.9$ |
| **Fasudil** | | | | | | | | |
| MAP (mmHg) | $86 \pm 9$ | $89 \pm 9$ | $92 \pm 9$ | $98 \pm 14$ | $93 \pm 11*$ | $96 \pm 12*$ | $99 \pm 12*$ | $104 \pm 15*$ |
| HR (beats min$^{-1}$) | $56 \pm 9$ | $63 \pm 8$ | $64 \pm 9$ | $70 \pm 16$ | $55 \pm 6$ | $58 \pm 7$ | $60 \pm 7$ | $63 \pm 10$ |
| $P_{vO_2}$ (mmHg) | $31.2 \pm 8.9*$ | $22.0 \pm 4.4$ | $22.5 \pm 4.0$ | $25.5 \pm 4.4$ | $27.3 \pm 4.0$ | $19.9 \pm 4.7$ | $21.7 \pm 3.0$ | $26.2 \pm 5.1$ |
| $P_{vCO_2}$ (mmHg) | $43.6 \pm 4.4$ | $46.6 \pm 4.0$ | $52.9 \pm 6.2$ | $56.3 \pm 9.2$ | $44.6 \pm 4.7$ | $47.9 \pm 5.6$ | $52.4 \pm 7.9$ | $51.6 \pm 8.4$ |
| a-vO$_2$ (ml l$^{-1}$) | $90.5 \pm 35.2*$ | $128.6 \pm 15.9$ | $134.0 \pm 13.7$ | $128.3 \pm 15.8$ | $98.8 \pm 20.4$ | $131.5 \pm 23.6$ | $126.8 \pm 16.4$ | $110.1 \pm 24.3$ |
| [Hb]$_V$ (g dl$^{-1}$) | $14.5 \pm 1.3$ | $14.4 \pm 1.4$ | $14.5 \pm 1.4$ | $14.7 \pm 1.4$ | $14.1 \pm 1.0$ | $13.9 \pm 1.1$ | $14.0 \pm 1.1$ | $14.0 \pm 0.7$ |

$*P < 0.05$ *vs.* saline.
$^{\dagger}P < 0.05$ *vs.* young (within condition).
Abbreviations: a-vO$_2$, arteriovenous oxygen difference; [Hb]$_V$, venous haemoglobin concentration; HR, heart rate; MAP, mean arterial pressure; $P_{vCO_2}$, venous partial pressure of carbon dioxide; $P_{vO_2}$, venous partial pressure of oxygen; $S_{pO_2}$, peripheral oxygen saturation.

exercise in older adults at 25% MVC *vs.* saline ($\Delta$FBF $304.2 \pm 90.9$ *vs.* $255.3 \pm 71.6$ ml min$^{-1}$ and $\Delta$FVC $289.6 \pm 74.5$ *vs.* $228.8 \pm 71.3$ ml min$^{-1}$ 100 mmHg$^{-1}$, respectively; $P = 0.001$ and $<0.0001$). In contrast, fasudil blunted the FBF and FVC responses in young adults at 25% MVC *vs.* saline ($\Delta$FBF $283.0 \pm 87.8$ *vs.* $315.4 \pm 107.3$ ml min$^{-1}$ and $\Delta$FVC: $283.5 \pm 70.3$ *vs.* $329.9 \pm 87.1$ ml min$^{-1}$ 100 mmHg$^{-1}$, respectively; $P = 0.038$ and $0.002$), and thus there was no longer an age-related impairment in haemodynamics (Fig. 3*A*–*D*).

Venous plasma [ATP] and ATP effluent at rest were not different between older and young adults in saline ($53.6 \pm 19.4$ *vs.* $74.1 \pm 41.1$ nmol l$^{-1}$ and $1.3 \pm 0.8$ *vs.* $1.6 \pm 1.1$ nmol min$^{-1}$, respectively; $P = 0.395$ and $0.967$) or fasudil ($74.3 \pm 46.1$ *vs.* $69.5 \pm 39.3$ nmol l$^{-1}$ and $2.0 \pm 1.4$ *vs.* $1.7 \pm 1.2$ nmo min$^{-1}$, respectively; $P = 0.842$ and $0.963$) conditions, and within each age group these values were not different between saline and fasudil ($P = 0.318–0.982$) (Fig. 3*E* and *F*). In the saline condition, plasma [ATP]$_V$ at 5% MVC was significantly lower in older vs young adults ($59.3 \pm 28.7$ *vs.* $107.7 \pm 52.3$ nmol l$^{-1}$; $P = 0.048$) and ATP effluent was lower at 25% MVC in older adults ($23.7 \pm 14.0$ *vs.* $43.6 \pm 31.0$ nmol min$^{-1}$; $P = 0.006$) (Fig. 3*E* and *F*). In older adults, fasudil increased plasma [ATP]$_V$ *vs.* saline at 5% ($128.0 \pm 65.1$ *vs.* $59.3 \pm 28.7$ nmol l$^{-1}$; $P = 0.010$) and 25% ($120.5 \pm 75.5$ *vs.* $82.0 \pm 38.8$ nmol l$^{-1}$; $P = 0.067$) MVC and significantly increased ATP effluent *vs.* saline at 25% MVC ($39.3 \pm 24.8$ *vs.* $23.7 \pm 14.0$ nmol min$^{-1}$; $P = 0.005$) such that there was no longer an age-related impairment (Fig. 3*E* and *F*). Sample size variability

between haemodynamics and plasma ATP measures is due to loss of catheter patency in some participants.

## Effects of age and fasudil on oxygen delivery and consumption during systemic isocapnic hypoxia and graded-intensity rhythmic handgrip exercise

In older adults, forearm oxygen delivery during hypoxia increased with fasudil *vs.* saline ($4.3 \pm 1.7$ *vs.* $3.5 \pm 1.3$ ml min$^{-1}$, $P = 0.053$) (Fig. 4*A*). During 25% MVC exercise, forearm oxygen delivery and $\dot{V}_{O_2}$ were lower in older *vs.* young adults with saline ($51.6 \pm 15.2$ *vs.* $69.9 \pm 28.6$ ml min$^{-1}$, $P = 0.005$, and $32.9 \pm 10.5$ *vs.* $42.4 \pm 15.0$ ml min$^{-1}$; $P = 0.016$, respectively), but not with fasudil ($58.4 \pm 16.8$ *vs.* $63.1 \pm 24.0$ ml min$^{-1}$, $P = 0.540$ ($P = 0.071$ *vs.* saline), and $34.8 \pm 11.7$ *vs.* $39.9 \pm 14.1$ ml min$^{-1}$; $P = 0.213$, respectively) (Fig. 4*B* and *F*). Sample size variability between haemodynamics and blood gas measures is due to loss of catheter patency in some participants.

## Effects of age and fasudil on isolated red blood cell extracellular and intracellular ATP

Blood gases for isolated RBCs are shown in Table 4. There were no age group or drug condition differences in the fraction of oxygenated haemoglobin ($F_{O_2Hb}$), an index of the stimulus for ATP release based on the linear relationship between haemoglobin oxygenation state and extracellular ATP. In contrast to young adults,

extracellular ATP from RBCs of older adults was not elevated in hypoxia compared to normoxia with saline ($14.1 \pm 8.9$ *vs.* $12.3 \pm 6.11.8$ nmol/$4 \times 10^8$ RBCs, respectively; $P = 0.214$) whereas it was with fasudil ($16.5 \pm 9.3$ *vs.* $11.6 \pm 6.8$ nmol/$4 \times 10^8$ RBCs, respectively; $P = 0.001$) (Fig. 5*A*). The RBC extracellular ATP response from normoxia to hypoxia was significantly lower in older

*vs.* young adults with saline ($15.0 \pm 50.5$ *vs.* $92.7 \pm 52.4\%$, respectively; $P = 0.001$) and fasudil ($53.0 \pm 46.6$ *vs.* $105.4 \pm 54.1\%$, respectively; $P$ 0.018), although there was a trend for fasudil to improve this change in extracellular ATP in the older adults relative to saline ($53.0 \pm 46.6$ *vs.* $15.0 \pm 50.5\%$, respectively; $P = 0.082$) (Fig. 5*B*). Finally, intracellular ATP increased in hypoxia ($P = 0.0001–0.002$)

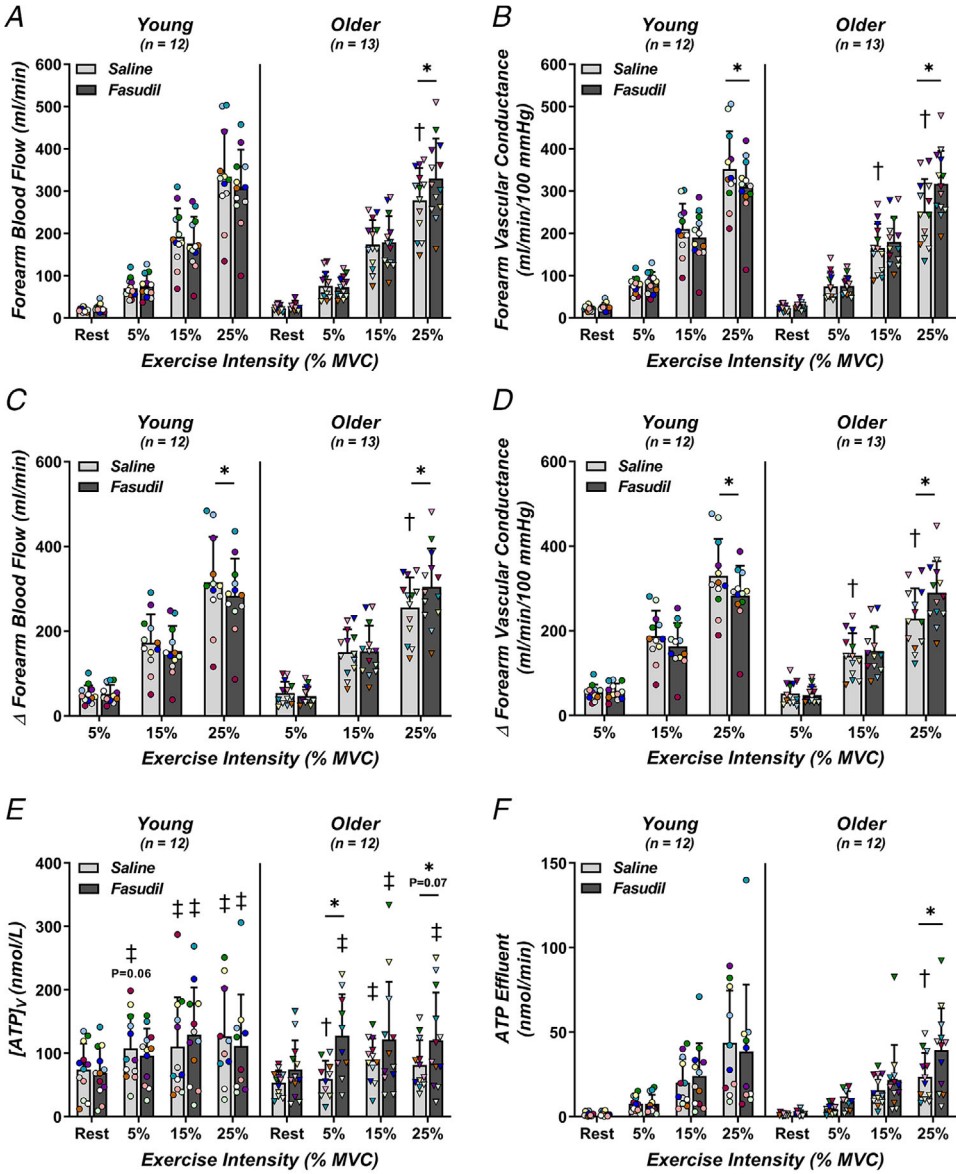

**Figure 3. Effects of age and fasudil on forearm haemodynamics and plasma ATP during rhythmic handgrip exercise**

*A* and *B*, with saline, forearm blood flow (FBF) was impaired during 25% maximum voluntary contraction (MVC) and forearm vascular conductance (FVC) was impaired during 15% and 25% MVC in older adults. Fasudil significantly increased FBF and FVC at 25% MVC in older adults and decreased FVC at 25% MVC in young adults. *C* and *D*, ΔFBF and ΔFVC responses from rest to exercise were impaired in older *vs.* young adults at 25% MVC with saline and were improved with fasudil in older adults, but not younger. *E* and *F*, with saline, [ATP]$_V$ and ATP effluent were significantly reduced in older *vs.* young adults at 5% and 25% MVC, respectively, and improved with fasudil at these same exercise intensities in older adults, but not younger. *P < 0.05 *vs.* saline; †*P* < 0.05 *vs.* young; ‡*P* < 0.05 *vs.* rest ([ATP]$_V$ only).

and there were no differences between age groups or drug conditions (Fig. 5C).

## Discussion

To the best of our knowledge, this is the first study to investigate the effects of Rho-kinase inhibition in healthy older adult humans as a potential means to improve age-related impairments in the control of vascular tone

and skeletal muscle blood flow during the physiological stimuli of hypoxia and exercise. The primary novel findings are as follows. First, fasudil improved the blunted local vasodilatory and blood flow responses to systemic hypoxia and moderate intensity (25% MVC) rhythmic handgrip exercise in older adults. Second, the haemodynamic improvements during systemic hypoxia and exercise with fasudil were associated with increased oxygen delivery to skeletal muscle, and thus reduced $\dot{V}_{O_2}$

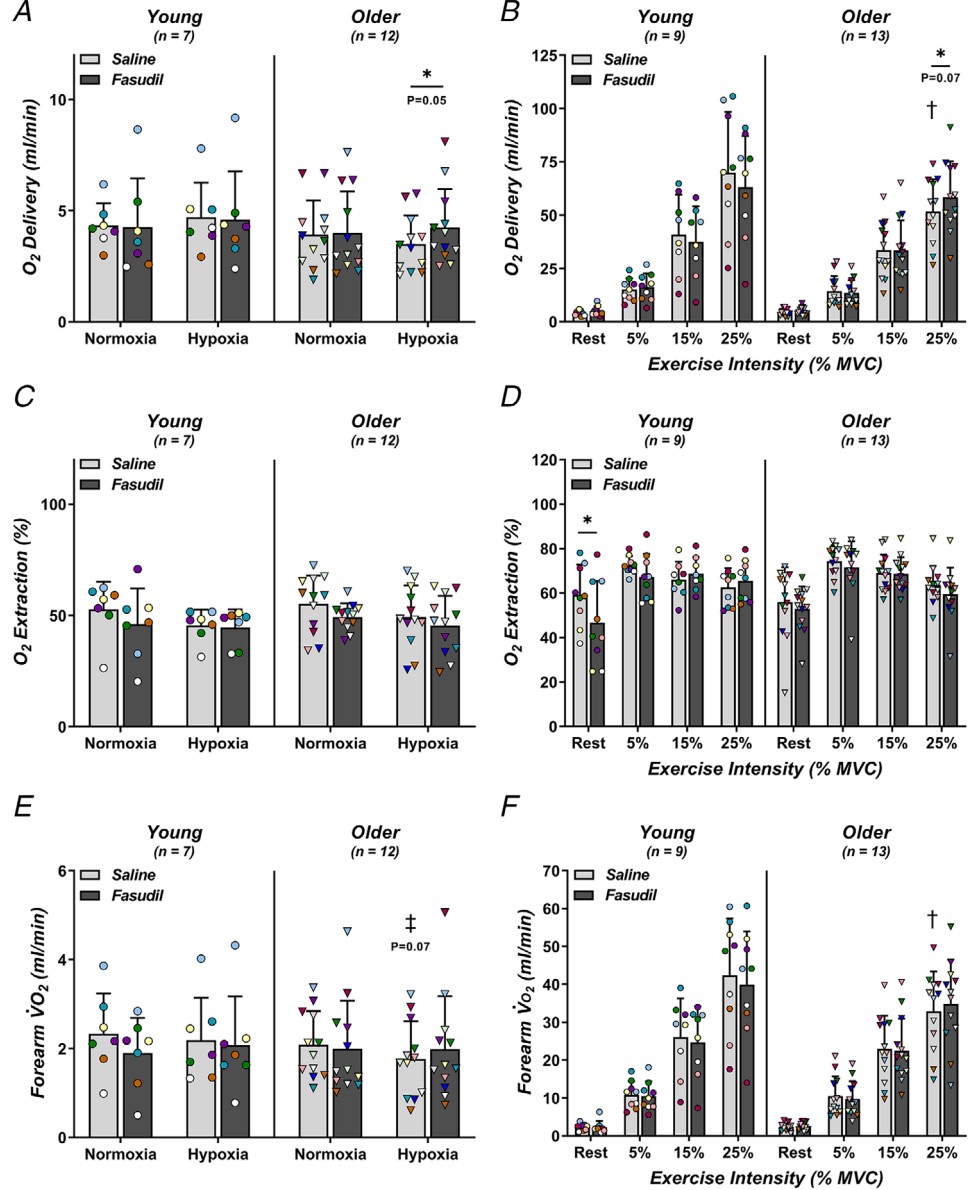

**Figure 4. Effects of age and fasudil on forearm oxygen delivery, extraction and consumption during hypoxia (*A*, *C* and *E*) and rhythmic handgrip exercise (*B*, *D* and *F*)**

*A*, *C* and *E*, there were no differences in forearm $O_2$ delivery (*A*), extraction (*C*), or consumption (*E*) between young and older adults. $\dot{V}_{O_2}$ tended to decrease during hypoxia in older adults with saline (*E*) and $O_2$ delivery was improved with fasudil (*A*). *B*, *D* and *F*, at 25% maximum voluntary contraction (MVC), forearm $O_2$ delivery (*B*) and $\dot{V}_{O_2}$ (*F*) were impaired in older *vs.* young adults with saline, which was no longer different with fasudil. *\*P < 0.05 vs.* saline; †*P < 0.05 vs.* young; ‡*P < 0.05 vs.* normoxia.

**Table 4. Isolated red blood cell gases**

| | | pH | $P_{O_2}$ (mmHg) | $P_{CO_2}$ (mmHg) | tHb (g dl$^{-1}$) | $F_{O_2 Hb}$ (%) | $F_{HHb}$ (%) |
|---|---|---|---|---|---|---|---|
| **Normoxia** | | | | | | | |
| Young | Saline | $7.42 \pm 0.05$ | $121.5 \pm 7.8$ | $33.9 \pm 2.1$ | $6.3 \pm 0.4$ | $95.5 \pm 0.6$ | $2.9 \pm 0.3$ |
| | Fasudil | $7.44 \pm 0.07$ | $123.0 \pm 12.7$ | $33.8 \pm 7.8$ | $6.3 \pm 0.5$ | $95.8 \pm 0.5$ | $3.0 \pm 0.4$ |
| Older | Saline | $7.39 \pm 0.06$ | $123.7 \pm 8.0$ | $38.1 \pm 4.2$ | $6.7 \pm 0.4^{†}$ | $95.5 \pm 0.4$ | $3.0 \pm 0.3$ |
| | Fasudil | $7.41 \pm 0.03$ | $126.4 \pm 5.5$ | $36.2 \pm 1.9$ | $6.7 \pm 0.4^{†}$ | $95.7 \pm 0.4$ | $2.8 \pm 0.4$ |
| **Hypoxia** | | | | | | | |
| Young | Saline | $7.44 \pm 0.02$ | $22.6 \pm 3.3$ | $35.4 \pm 2.4$ | $6.3 \pm 0.5$ | $34.1 \pm 9.8$ | $62.2 \pm 9.6$ |
| | Fasudil | $7.45 \pm 0.01$ | $21.3 \pm 3.1$ | $34.8 \pm 2.2$ | $6.3 \pm 0.4$ | $32.1 \pm 9.6$ | $64.5 \pm 8.7$ |
| Older | Saline | $7.41 \pm 0.04$ | $25.5 \pm 2.1$ | $39.2 \pm 2.6$ | $6.8 \pm 0.5^{†}$ | $37.9 \pm 4.8$ | $58.4 \pm 4.8$ |
| | Fasudil | $7.41 \pm 0.03$ | $24.8 \pm 3.0$ | $40.1 \pm 2.5^{†}$ | $6.7 \pm 0.5$ | $36.0 \pm 9.9$ | $60.5 \pm 9.6$ |

*$P < 0.05$ *vs*. saline.
$^{†}P < 0.05$ *vs*. young; $n = 11$ (young) and 12 (older). Abbreviations: $F_{HHb}$, fraction of deoxygenated haemoglobin; $F_{O_2 Hb}$, fraction of oxygenated haemoglobin; $P_{CO_2}$, partial pressure of carbon dioxide; $P_{O_2}$, partial pressure of oxygen; tHb, total haemoglobin.

was no longer observed with these physiological stimuli in older adults. Third, fasudil significantly lowered mean arterial blood pressure in healthy, normotensive older adults at rest, during systemic hypoxia, and during exercise such that age-related elevations in control conditions were abolished. Fourth, improvements in haemodynamic responses with fasudil were accompanied by improvements in circulating ATP in older adults based on trends for increased venous plasma [ATP] and significantly improved ATP effluent. Finally, *in vivo* fasudil administration tended to improve the age-related impairment in deoxygenation-induced ATP release from isolated RBCs. These collective findings provide the first experimental evidence that systemic Rho-kinase inhibition improves the regulation of vascular tone and skeletal muscle blood flow, along with circulating ATP responses, during the physiological stimuli of systemic hypoxia and moderate intensity handgrip exercise in healthy older adults, which may have therapeutic implications for reducing cardiovascular disease risk and increasing exercise tolerance and functional capacity in ageing populations.

### Ageing and impaired vascular control during hypoxia and exercise

The local regulation of blood flow involves the integration of multiple signalling pathways and vascular responses, the end goal of which is the matching of oxygen supply to tissue metabolic demand. Changes in these pathways with advancing age, particularly an attenuation of local vasodilatory signalling, contribute to age-related impairments in vascular control during physiological stimuli such as hypoxia (Casey et al., 2011; Kirby et al., 2012; Richards et al., 2017) and exercise (Hearon Jr. & Dinenno, 2016; Proctor & Parker, 2006; Wray & Richardson, 2015).

Data from the present study are consist with these prior observations and demonstrate impaired blood flow responses due to impaired control of vascular tone in older adults. RBCs may play a central role in the coupling of oxygen supply and metabolic demand through the release of ATP in direct proportion to the degree of haemoglobin deoxygenation (Bergfeld & Forrester, 1992; Dietrich et al., 2000; Ellsworth, 2000; Ellsworth & Sprague, 2012; Ellsworth et al., 1995; González-Alonso et al., 2002; Jagger et al., 2001; Jensen, 2009; Sprague et al., 2009), and interestingly, this process is also impaired with advancing age (Kirby et al., 2012; Racine & Dinenno, 2019). We recently demonstrated that treatment of isolated RBCs from older adults with a rho-kinase inhibitor *in vitro* significantly improved ATP release during haemoglobin deoxygenation (Racine & Dinenno, 2019), thus serving as the impetus for the present study.

### Effects of Rho-kinase inhibition on vascular control in older adults

In this first study, to the best of our knowledge, of whether systemic Rho-kinase inhibition can improve vascular and haemodynamic responses to hypoxia and exercise in young or older adults, fasudil completely reversed the age-related impairment in hypoxic vasodilation, resulting in improved blood flow and oxygen delivery. Further, fasudil significantly improved forearm vasodilatation and hyperaemia during moderate intensity handgrip exercise, resulting in a greater oxygen delivery and forearm $\dot{V}_{O_2}$ that was no longer significantly lower with age compared with control conditions. In contrast and unexpectedly, fasudil blunted forearm vasodilator and hyperaemic responses during high-intensity exercise in the young adults, and as such, the age group differences in forearm haemodynamics and oxygen consumption were

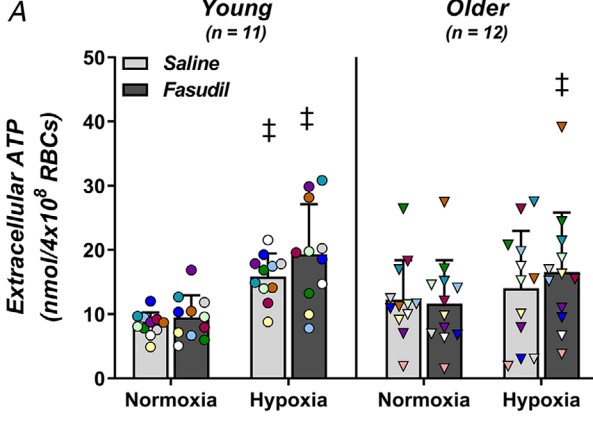

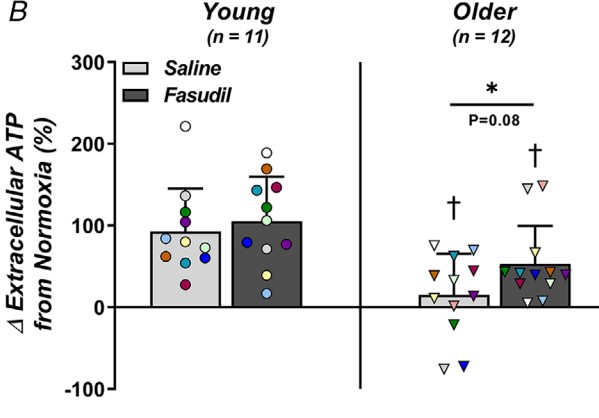

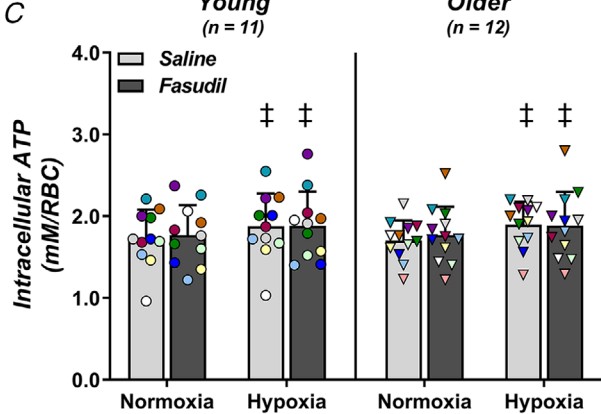

**Figure 5. Effects of donor age and *in vivo* fasudil administration on red blood cell ATP release and intracellular ATP in normoxia and hypoxia**

*A*, extracellular ATP (i.e. ATP release) in hypoxia only increased significantly *vs.* normoxia from RBCs of older adults with fasudil, whereas it increased from RBCs of young adults in both conditions. *B*, the mean percentage increase in extracellular ATP from normoxia to hypoxia was impaired from RBCs of older adults in the saline and fasudil conditions, but tended to improve with fasudil compared to saline in older adults ($P = 0.08$). *C*, intracellular ATP increased from normoxia to hypoxia in all conditions and there were no differences between young and older adults. *$P < 0.05$ *vs.* saline; †$P < 0.05$ *vs.* young; ‡$P < 0.05$ *vs.* normoxia.

abolished. Given that we estimated $C_{aO_2}$ in the present study and that blood gasses were not obtained in all subjects, caution should be taken when interpreting the findings related to forearm $\dot{V}_{O_2}$.

### Possible mechanisms of fasudil-mediated improvements in vascular control in older adults

Our laboratory has demonstrated previously that age-related reductions in skeletal muscle blood flow during conditions of RBC deoxygenation are due to impaired local vasodilatation associated with reduced circulating ATP (Kirby et al., 2012), that increases in circulating ATP are dependent on skeletal muscle perfusion and thus an intravascular source (Kirby et al., 2013), and that impaired deoxygenation-induced ATP release from isolated RBCs of older adults is mechanistically linked to declines in RBC deformability with advancing donor age (Racine & Dinenno, 2019). Importantly, we also demonstrated that improved RBC deformability with advancing donor age via Rho-kinase inhibition acutely increases RBC ATP release (Racine & Dinenno, 2019). Although the design of the present study does not permit identification of the precise underlying mechanisms of fasudil-mediated improvements in vaso-dilatory and blood flow responses to systemic hypoxia and moderate intensity rhythmic handgrip exercise in older adults, there were multiple significant or trending improvements in circulating ATP during these stimuli when participants were treated with fasudil compared to saline (Figs 2*E*, *F* and 3*E*, *F*) in addition to a trend for improved isolated RBC ATP release following *in vivo* drug treatment (Fig. 5*B*). Given the ability of ATP to stimulate both local and conducted vasodilatation (Dora, 2017; Winter & Dora, 2007) and the preserved vaso-dilatory responsiveness to exogenous ATP in the forearm of older adults (Kirby et al., 2010), these data suggest that haemodynamic improvements with fasudil during conditions of RBC deoxygenation may be due at least in part to increased ATP bioavailability.

Given the diverse molecular targets of Rho-kinase *in vivo* and the systemic administration of fasudil in the present study, there are multiple pathways beyond ATP signalling that could have contributed to the fasudil-mediated improvements in vascular control in older adults. Rho-kinase inhibits the synthesis of nitric oxide (NO) by phosphorylating endothelial nitric oxide synthase (eNOS) at threonine 495 (Satoh et al., 2014; Shimokawa et al., 2016; Sugimoto et al., 2007). In healthy humans, intra-arterial infusion of fasudil has been shown to decrease forearm vascular tone, mediated in part by improved NO bioavailability as a result of decreased inhibition of eNOS (Büssemaker et al., 2007). Although we did not observe an increase in resting FVC in young or older adults when treated with systemic fasudil in the

present study (Figs 2*B* and 3*B*), mean arterial pressure was significantly reduced with fasudil *vs.* saline in older adults (Tables 2 and 3). Given that heart rate was not different between study treatments (Tables 2 and 3), this suggests that the decrease in MAP was due to a decrease in systemic vascular resistance, which could be due in part to increased systemic NO bioavailability given the contribution of NO to basal vascular tone (Vallance et al., 1989). If NO bioavailability was increased by fasudil in the present study, it is possible that it contributed to the improved forearm vasodilatation during systemic hypoxia (Fig. 2; Blitzer et al., 1996; Casey et al., 2011; Crecelius et al., 2011; Markwald et al., 2011), as well as during moderate intensity rhythmic handgrip exercise (Crecelius et al., 2010; Kirby et al., 2009; Richards et al., 2015), in older adults.

Beyond effects on vasodilatory pathways, the Rho-kinase inhibitors fasudil and Y-27632 have both been shown to blunt the vascular response to multiple vasoconstrictors. Specifically, fasudil can limit $\alpha$-adrenergic vasoconstriction induced by noradrenaline and abolish endothelin-mediated vasoconstriction (Büssemaker et al., 2007) and Y-27632 has been shown to cause a dose-dependent decrease in $\alpha_1$-adrenergic vasoconstriction stimulated by phenylephrine (Löhn et al., 2005). If some degree of inhibition of $\alpha$-adrenergic vasoconstriction occurred in the present study, it is unlikely that it contributed to the improved vascular responses in older adults given our previous observations that local adrenoceptor blockade does not improve the age-related impairments in forearm vasodilatation during hypoxia (80% $S_{pO_2}$) or graded-intensity rhythmic handgrip exercise (Richards et al., 2014, 2017). However, it is possible that the effect of fasudil on endothelin-mediated constriction (Büssemaker et al., 2007) could have contributed in the present study given that an age-related increase in vasoconstriction mediated by endothelin A receptors has been identified in the leg of healthy older adults both at rest and during exercise (Barrett-O'Keefe et al., 2015).

### Experimental limitations

The primary experimental limitation of the present study is that fasudil could not be delivered specifically to RBCs and therefore systemic drug administration was necessary to attempt to inhibit Rho-kinase in the circulating pool of RBCs. Given that Rho-kinase is distributed widely throughout the body and has a diverse range of cellular targets, follow-up studies will be needed to gain more definitive mechanistic insight into the fasudil-mediated improvements in vascular control in healthy older adults. Utilizing systemic fasudil administration in combination with local brachial artery infusion of pharmacological antagonists targeting pathways that could be altered

by Rho-kinase inhibition as described above would provide insight into the mechanisms contributing to fasudil-mediated haemodynamic improvements (Barrett-O'Keefe et al., 2015; Crecelius et al., 2012; Racine et al., 2018).

Another limitation relates to the inherent difficulties of *in vivo* plasma/circulating ATP measures. In this context, ATP in circulation has an extremely short half-life ($\sim$0.5 s; Mortensen et al., 2011) and we are unable to measure ATP within the microcirculation. As such, this likely contributes to the mixed/variable data as it relates to both age-impairments and fasudil-mediated improvements in plasma [ATP]. In this study, it was also not possible to measure RBC deformability as in our previous *in vitro* study utilizing Rho-kinase inhibition (Racine & Dinenno, 2019) due to time and personnel constraints. Thus, it is possible that systemic Rho-kinase inhibition *in vivo* did not improve RBC deformability. However, it clearly improved forearm vasodilator and haemodynamic responses to hypoxia and moderate intensity handgrip exercise in older adults.

A final limitation of systemic drug administration is that fasudil and its active metabolite hydroxyfasudil can have off-target effects from Rho-kinase, including myosin light chain kinase, protein kinase A and protein kinase C. However, the inhibitor constant ($K_i$) of fasudil is much more specific for Rho-kinase, ranging from 0.33 to 1.9 $\mu$M (Davies et al., 2000; Jacobs et al., 2006; Rikitake et al., 2005; Satoh et al., 2012; Shibuya et al., 2005; Wickman et al., 2003) compared to 55 $\mu$M for myosin light chain kinase (Satoh et al., 2012), $\sim$10 $\mu$M on average for protein kinase A (Davies et al., 2000; Jacobs et al., 2006; Rikitake et al., 2005; Satoh et al., 2012), and ranging from 3.3 $\mu$M to over 100 $\mu$M for protein kinase C (Rikitake et al., 2005; Satoh et al., 2012). Importantly, hydroxyfasudil is an even more potent and specific inhibitor of Rho-kinase than fasudil, with a $K_i$ ranging from 0.039 to 1.8 $\mu$M (Jacobs et al., 2006; Rikitake et al., 2005; Satoh et al., 2012; Shibuya et al., 2005; Shimokawa, 2002; Shimokawa et al., 1999) compared to 140 $\mu$M for myosin light chain kinase (Satoh et al., 2012), 2.2–37 $\mu$M for protein kinase A (Jacobs et al., 2006; Rikitake et al., 2005; Satoh et al., 2012) and 18−100 $\mu$M for protein kinase C (Rikitake et al., 2005; Satoh et al., 2012; Shimokawa et al., 1999), and it has a significantly longer half-life in circulation compared to fasudil (over 4 h *vs.* less than 1 h) (Shibuya et al., 2005). Given that plasma [fasudil] and [hydroxyfasudil] in the present study were within the $K_i$ range specific for Rho-kinase, it is unlikely that any effects of fasudil in the present study were due to off-target effects of these compounds.

### Experimental considerations

Given our study design in which the order of saline and fasudil study days were randomized, and that there is

some day-to-day variability in forearm haemodynamics, absolute levels of FBF and FVC were not always significant between groups. This was observed during the hypoxia trial, but when the data are expressed as a response from baseline (i.e. Δ FBF or FVC) to account for individual differences in baseline haemodynamics, there is a clear age-related impairment in hypoxic vasodilatation that is reversed with fasudil (Fig. 2). During graded handgrip exercise, a similar observation was made during 15% MVC exercise, but a reduction in FBF and FVC was observed during 25% MVC exercise in older adults whether expressed as absolute levels or a change from baseline (Fig. 3). Finally, independent of how the forearm haemodynamic data are expressed, Rho-kinase inhibition with fasudil clearly improved the responses to hypoxia and moderate intensity handgrip exercise in older adults only.

Another important experimental consideration is that the effect of Rho-kinase inhibition on vascular tone and blood flow in older adults during exercise was specific to moderate intensity rhythmic handgrip exercise. Additionally, the age group difference in absolute levels of muscle blood flow and vascular conductance during exercise was ~18% and 29%, respectively, consistent with prior studies on this topic (for review see Hearon Jr. & Dinenno, 2016). Future studies will be required to determine whether Rho-kinase inhibition improves exercising muscle haemodynamics during larger muscle mass (e.g. knee extensor) or more conventional exercise modalities (e.g. cycling), and whether any increase in muscle blood flow and vascular conductance results in improved exercise capacity or tolerance, to ascertain the therapeutic potential of pharmacologically induced Rho-kinase inhibition in older and/or diseased humans.

## Conclusions

This investigation provides the first experimental evidence that age-related impairments in the peripheral vasodilatory and hyperaemic responses to systemic hypoxia and moderate intensity rhythmic handgrip exercise can be significantly improved by Rho-kinase inhibition in humans, and that this is accompanied by improvements in circulating ATP during these physiological stimuli. These findings also provide the necessary foundation for performing future investigations to determine the underlying mechanisms by which Rho-kinase inhibition improves these haemodynamic responses in healthy older adults. Thus, Rho-kinase inhibition and/or targeting RBC ATP release may be a promising therapeutic target for improving blood flow and oxygen delivery in older adults, and for ultimately ameliorating the age-related declines in exercise tolerance and functional independence associated with impaired vascular function with advancing age and various patient populations.

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

## Additional information

### Data availability statement

All data supporting the results presented in the manuscript are included in the manuscript figures ($n \leq 30$), as per *The Journal*'s Statistics Policy.

### Competing interests

None.

## Author contributions

All experiments were performed in the Human Cardiovascular Physiology Laboratory, Colorado State University, Fort Collins, CO, USA. M.L.R., J.D.T., N.B.K., N.P.B., J.C.R. and F.A.D. contributed to conception and design of the experiments, data collection, analysis, and interpretation, and writing/revising the manuscript. G.J.L. contributed to the experimental design, data collection, and critical revision of the manuscript. All authors approved the final version of the manuscript and agree to be accountable for all aspects of the work. All persons designated as authors qualify for authorship, and all those who qualify for authorship are listed.

## Funding

This research was supported by the National Institutes of Health awards HL119337 (F.A.D., M.J.J.) and F31HL126377 (M.L.R., F.A.D.).

## Acknowledgements

We thank the subjects who volunteered to participate and all laboratory volunteers for their assistance in conducting these studies.

## Keywords

ageing, ATP, blood flow, exercise, fasudil, hypoxia

## Supporting information

Additional supporting information can be found online in the Supporting Information section at the end of the HTML view of the article. Supporting information files available:

**Peer Review History**
**Statistical Summary Document**

