## [Peer Review History · The Journal of Physiology]

Rho-kinase inhibition improves haemodynamic responses & circulating ATP during hypoxia & moderate intensity handgrip exercise in healthy older adults

Matthew L. Racine, Janée D. Terwoord, Nathaniel B. Ketelhut, Nate P. Bachman, Jennifer C. Richards, Gary J Luckasen, and Frank A. Dinunno

DOI: 10.1113/JP282730

Corresponding author(s): Frank Dinunno (frank.dinunno@colostate.edu)

Review Timeline:

Submission Date:	12-Dec-2021
Editorial Decision:	25-Jan-2022
Revision Received:	31-Mar-2022
Editorial Decision:	14-Apr-2022
Revision Received:	20-Apr-2022
Editorial Decision:	28-Apr-2022
Revision Received:	28-Apr-2022
Accepted:	09-May-2022

Senior Editor: Michael Hogan

Reviewing Editor: Bruno Grassi

Transaction Report:

Dear Dr Dinunno,

Re: JP-RP-2021-282730 "Rho-kinase inhibition improves haemodynamic responses and circulating ATP during hypoxia and exercise in healthy older adults" by Matthew L. Racine, Janée D. Terwoord, Nathaniel B. Ketelhut, Nate P. Bachman, Jennifer C. Richards, Gary J Luckasen, and Frank A. Dinunno

Thank you for submitting your manuscript to The Journal of Physiology. It has been assessed by a Reviewing Editor and by 2 expert Referees and I am pleased to tell you that it is considered to be acceptable for publication following satisfactory revision.

The reports are copied at the end of this email. Please address all of the points and incorporate all requested revisions, or explain in your Response to Referees why a change has not been made.

All raw data referred to in the manuscript must be provided.

If n {less than or equal to} 30, all data points must be plotted within the figure. We recommend authors use a box and whisker plot to present their data. Note: if each subject has numerous data points associated with it (e.g. time course data), we would treat n as being each data point, not the number of subjects.

If $n > 30$, then the entire raw dataset must be submitted as supporting information, or hosted on a not-for-profit repository e.g. FigShare, with access details provided in the Data Availability Statement in the article file.

The most appropriate summary statistic (e.g. mean or median and standard deviation) must be plotted.

A bar graph with data points overlaid or a box and whisker plot are acceptable formats.

Data summaries in the text should be presented as mean (SD) with a clear statement of n ; and presented in each key location:

PLEASE NOTE **Standard Deviation (SD) must be used rather than Standard Error of the Mean (SEM). Manuscripts that report only SEM will be returned to authors.

All relevant 'n' values must be clearly stated at each key location and 'n' defined (e.g. x cells from y slices in z animals) in the methods.

Exact p values must be stated. Authors must not use 'greater than' or 'less than'. This is essential so that others can assess the evidence for a conclusion.

Exact p values must be stated even when 'no statistical significance' is claimed.

Authors must refrain from making trend statements such as 'x increased, but was not significant'.

NEW POLICY: In order to improve the transparency of its peer review process The Journal of Physiology publishes online as supporting information the peer review history of all articles accepted for publication. Readers will have access to decision letters, including all Editors' comments and referee reports, for each version of the manuscript and any author responses to peer review comments. Referees can decide whether or not they wish to be named on the peer review history document.

Authors are asked to use The Journal's premium BioRender (<https://biorender.com/>) account to create/redrawn their Abstract Figures. Information on how to access The Journal's premium BioRender account is here: <https://physoc.onlinelibrary.wiley.com/journal/14697793/biorender-access> and authors are expected to use this service. This will enable Authors to download high-resolution versions of their figures.

I hope you will find the comments helpful and have no difficulty returning your revisions within 4 weeks.

Your revised manuscript should be submitted online using the links in Author Tasks Link Not Available.

Any image files uploaded with the previous version are retained on the system. Please ensure you replace or remove all files that have been revised.

REVISION CHECKLIST:

- Article file, including any tables and figure legends, must be in an editable format (eg Word)
- Abstract figure file (see above)
- Statistical Summary Document
- Upload each figure as a separate high quality file
- Upload a full Response to Referees, including a response to any Senior and Reviewing Editor Comments;
- Upload a copy of the manuscript with the changes highlighted.

- A potential 'Cover Art' file for consideration as the Issue's cover image;
- Appropriate Supporting Information (Video, audio or data set https://jp.msubmit.net/cgi-bin/main.plex?form_type=display_requirements#supp).

To create your 'Response to Referees' copy all the reports, including any comments from the Senior and Reviewing Editors, into a Word, or similar, file and respond to each point in colour or CAPITALS and upload this when you submit your revision.

I look forward to receiving your revised submission.

If you have any queries please reply to this email and staff will be happy to assist.

Yours sincerely,

Michael C. Hogan
 Senior Editor
 The Journal of Physiology
<https://jp.msubmit.net>
<http://jp.physoc.org>
 The Physiological Society
 Hodgkin Huxley House
 30 Farringdon Lane
 London, EC1R 3AW
 UK
<http://www.physoc.org>
<http://journals.physoc.org>

REQUIRED ITEMS:

-Author photo and profile. First (or joint first) authors are asked to provide a short biography (no more than 100 words for one author or 150 words in total for joint first authors) and a portrait photograph. These should be uploaded and clearly labelled with the revised version of the manuscript. See Information for Authors for further details.

-You must start the Methods section with a paragraph headed Ethical Approval. If experiments were conducted on humans confirmation that informed consent was obtained, preferably in writing, that the studies conformed to the standards set by the latest revision of the Declaration of Helsinki, and that the procedures were approved by a properly constituted ethics committee, which should be named, must be included in the article file. If the research study was registered (clause 35 of the Declaration of Helsinki) the registration database should be indicated, otherwise the lack of registration should be noted as an exception (e.g. The study conformed to the standards set by the Declaration of Helsinki, except for registration in a database.). For further information see: <https://physoc.onlinelibrary.wiley.com/hub/human-experiments>

-Please upload separate high-quality figure files via the submission form.

-Please ensure that the Article File you upload is a Word file.

-A Statistical Summary Document, summarising the statistics presented in the manuscript, is required upon revision. It must be on the Journal's template, which can be downloaded from the link in the Statistical Summary Document section here: https://jp.msubmit.net/cgi-bin/main.plex?form_type=display_requirements#statistics

-Papers must comply with the Statistics Policy https://jp.msubmit.net/cgi-bin/main.plex?form_type=display_requirements#statistics

In summary:

-If $n \leq 30$, all data points must be plotted in the figure in a way that reveals their range and distribution. A bar graph with data points overlaid, a box and whisker plot or a violin plot (preferably with data points included) are acceptable formats.

-If $n > 30$, then the entire raw dataset must be made available either as supporting information, or hosted on a not-for-profit repository e.g. FigShare, with access details provided in the manuscript.

- n clearly defined (e.g. x cells from y slices in z animals) in the Methods. Authors should be mindful of pseudoreplication.

-All relevant n values must be clearly stated in the main text, figures and tables, and the Statistical Summary Document (required upon revision)

-The most appropriate summary statistic (e.g. mean or median and standard deviation) must be used. Standard Error of the Mean (SEM) alone is not permitted.

-Exact p values must be stated. Authors must not use 'greater than' or 'less than'. Exact p values must be stated to three significant figures even when 'no statistical significance' is claimed.

-Statistics Summary Document completed appropriately upon revision

-A Data Availability Statement is required for all papers reporting original data. This must be in the Additional Information section of the manuscript itself. It must have the paragraph heading "Data Availability Statement". All data supporting the results in the paper must be either: in the paper itself; uploaded as Supporting Information for Online Publication; or archived in an appropriate public repository. The statement needs to describe the availability or the absence of shared data. Authors must include in their Statement: a link to the repository they have used, or a statement that it is available as Supporting Information; reference the data in the appropriate section(s) of their manuscript; and cite the data they have shared in the References section. Whenever possible the scripts and other artefacts used to generate the analyses presented in the paper should also be publicly archived. If sharing data compromises ethical standards or legal requirements then authors are not expected to share it, but must note this in their Statement. For more information, see our Statistics Policy.

-Please include an Abstract Figure. The Abstract Figure is a piece of artwork designed to give readers an immediate understanding of the research and should summarise the main conclusions. If possible, the image should be easily 'readable' from left to right or top to bottom. It should show the physiological relevance of the manuscript so readers can assess the importance and content of its findings. Abstract Figures should not merely recapitulate other figures in the manuscript. Please try to keep the diagram as simple as possible and without superfluous information that may distract from the main conclusion(s). Abstract Figures must be provided by authors no later than the revised manuscript stage and should be uploaded as a separate file during online submission labelled as File Type 'Abstract Figure'. Please ensure that you include the figure legend in the main article file. All Abstract Figures should be created using BioRender. Authors should use The Journal's premium BioRender account to export high-resolution images. Details on how to use and access the premium account are included as part of this email.

EDITOR COMMENTS

Reviewing Editor:

The study supports a role of RBC rho-kinase in plasma ATP and blood flow, and the potential of Rho-kinase inhibition as an intervention that increases forearm blood flow during intense exercise in old individuals. Data are novel, nicely presented and discussed, and the study could have a significant impact on the field. The research group has a solid experience in the field.

Both reviewers make precise comments and raise some issues (mainly related to data interpretation and analysis) which should be convincingly addressed by the authors.

REFEREE COMMENTS

Referee #1:

GENERAL COMMENTS

This comprehensive study employed the rhythmic handgrip exercise model in conjunction with in vivo measurements of blood flow, blood oxygenation and plasma ATP in young and older individuals in normoxia, hypoxia and venous fusadil infusion and ex-vivo RBC ATP release in comparable conditions to investigate the impact of RBC rho-kinase inhibition in limb blood flow and circulating ATP. Interestingly, no absolute differences in forearm blood flow (FBF) or plasma [ATP] were observed between the older and young groups at rest in normoxia, at rest in hypoxia or during most of the subpeak rhythmic handgrip exercise bouts with either saline or fusadil infusion. The main positive findings were that systemic venous fusadil infusion induced 1) increases (above baseline) in FBF in hypoxia in both groups and 2) elevations in peak exercise hyperaemia in the older group (compared to baseline), but not in the young individuals. No absolute between group differences in FBF were observed at peak exercise with fusadil infusion. The study appears to have been well-conducted and the results with fusadil infusion and ex-vivo RBC ATP release are novel. That being said, the paper overemphasises the positive results in the peak exercise condition and fails to provide the reader with information about the absolute differences in FBF as well as information about the conditions where FBF was not different between groups. This is important to put in perspective the magnitude of the phenomenon under investigation and thus provide insight into its therapeutic potential. There is also a need to clarify how arterial O₂ content and forearm VO₂ were estimated to convince the reader of the validity of the reported statistical differences in forearm VO₂, which are within the measurement error of the multiple variables used in this calculation. Lastly, it would be helpful to acknowledge in the experimental considerations and limitations section that further work is needed in older people undertaking other exercise modalities (including exercise capacity tests during larger muscle mass exercise) to ascertain the therapeutic potential of pharmacologically induced rho-kinase inhibition.

SPECIFIC COMMENTS

TITLE - Specify that the rhythmic handgrip exercise model was employed. The effects of Rho-kinase inhibition during other exercise modalities remains unknown and thus the results cannot be generalised.

KEY POINTS - the background information tells the reader that a general impairment in skeletal muscle blood flow occurs with exposure to hypoxia and exercise, but this is not supported by the data of the present study, which only showed FBF differences at peak exercise in the saline infusion trial. The general statements in the first and last key points therefore needs to be amended to specify the conditions where blood flow in older people is lower and thus its regulation might be 'impaired' (or altered) compared to young individuals.

ABSTRACT - State whether the absolute exercise intensities were comparable between groups. As the authors know, this is critical to determine hemodynamic differences between groups, as the metabolic demand determined by the absolute work rate influences the magnitude of blood flow. Similarly, provide quantitative data of the magnitude of blood flow differences between groups to establish the physiological significance of the phenomenon.

ABSTRACT - Is the conclusion relating the improved circulating ATP and blood flow in older group based on any correlational analysis of FBF vs. plasma [ATP]?

INTRODUCTION - End of the introduction. Clarify why the rhythmic handgrip exercise is used to answer the question of the study.

METHODS - Page 10. The absolute forearm vascular conductance values in this model are microliters/min/mmHg. Why do they need to be expressed per 100 mmHg?

Page 10. The authors report the weights corresponding to the workloads used in the study in Table 1. Did you also measure the rate of contraction to obtain some index of power? The reported weight data do not seem to indicate that power was the same in both groups, as indicated by the statistical analysis.

Page 14. The estimates of arterial O₂ content are not clearly explained. Why should they be the same for all participants during exposure to normoxia (203 ml/l) and hypoxia (165 ml/l)? This has a bearing in the calculations of O₂ delivery, O₂ extraction and VO₂, particularly when absolute values are relatively small compared to other exercise and environmental conditions. Please explain.

RESULTS - Page 15. See comment above about workloads. Although 0.1, 0.5 and 0.7 kg 'lower' mean absolute workloads in the older group are not 'statistically' different from the young group, they can still affect blood flow values. See below the estimate of peak FBF per kg.

Page 16. Consider changing the increase in FBF and FVC was 'impaired' for 'was lower'.

Page 16. The magnitude of the phenomenon seems small when considering that forearm vascular conductance values were 9 vs. 6 microliters/min/mmHg.

Page 16. This reviewer is not sure about the importance of the ATP effluent data when knowing that venous ATP infusion do not change limb flow. The level of plasma [ATP] is more relevant to explain blood flow changes in the present conditions. Blood flow is a confounding variable when relating changes in ATP effluent to changes in blood flow.

Page 17. Avoid using the word 'impaired' in the results section. This is an interpretation of the data, which is more suited for the discussion section. Here you could simply state whether data were higher, lower, or no different.

Page 17. Consider normalising the data for the workload, as part of the differences in FBF, O₂ delivery and VO₂ can be accounted for by the lower absolute workload (0.7 kg at peak).

TABLES & FIGURES. There is duplication in the data presentation. To remedy this, figures could include not only the mean {plus minus} SD but also the individual data. The graphs with the delta data are not needed.

TABLES - Directly measured CvO₂ and [Hb] values could be reported below FBF and FVC data. The values of PvO₂ and PvCO₂ would also be helpful to know whether the experimental conditions of the ex-vivo experiments were comparable to those in the vivo experiment.

FIGURE 4. A graph illustrating the forearm a-vO₂ difference would be more insightful than O₂ extraction in discerning the factors accounting for the small differences in forearm VO₂ in some conditions.

DISCUSSION - Summary and other paragraphs interpreting the data. The conclusion paragraph reflects the findings, but do not provide an idea about the magnitude of the phenomenon under investigation, or make it clear to the reader that

differences in FBF were only observed at peak exercise in the saline trial. This is crucial to be able to evaluate the potential implications of the study for older people undertaking other ecologically relevant exercise modalities. Statements of the magnitude of difference in FBF between older and young people during exercise and the variability in this and plasma [ATP] measurements are needed to address this concern.

There is little discussion of the negative observations revealing similar between groups FBF at rest (normoxia and hypoxia) or during handgrip exercise at 5%MVC and 15%MVC with either saline or fasudil infusions. Between group FBF differences were only observed at 25% MVC (335 vs. 278 mL/min in older; $P < 0.05$). This effect during the saline condition could be in part confounded by differences in workload (8.1 vs 7.4 kg; FBF per kg = 41 vs. 38 ml/min/kg in young vs. older group). Please clarify whether the normalised data would still be statistically different.

Referee #2:

The manuscript by Racine et al describes potential mechanisms to ameliorate age-induced impairments in skeletal muscle vascular function and hemodynamic responses through improvements in circulating ATP during both hypoxia and hand grip exercise. The authors demonstrate that decreased ATP release may be involved in the increased risk of CVD and the decline in functional capacity with primary aging. Further, the use of Fasudil may alleviate age-associated CVD risk and/or the declines in functional capacity. The experimental paradigm is well designed, and the manuscript is well written. However, there are some concerns with the data and analysis and aspects of the discussion that must be addressed. Below are major and minor concerns.

MAJOR CONCERNS

1. In the key points section, you state "...fasudil, improved blood flow and circulating ATP responses during hypoxia and hand grip exercise compared to adults...". However, in Figure 2, this does not appear to be the case during hypoxia. Unless I am missing something, there is no comparison made between older adults (OA) and young adults (YA).

2. In figure 2E, OA [ATP]_v with fasudil and hypoxia is significantly different versus fasudil and normoxia, however, in table 3, they are not significantly different. With hypoxia and fasudil versus hypoxia and saline in OA, [ATP]_v is not different. Further, in figure 2F, hypoxia and fasudil did not increase ATP effluent versus saline and hypoxia in OA. Please explain these discrepancies and lack of significant difference with Fasudil versus saline, as this is clearly a primary outcome.

3. Figure 3A-D: How was delta blood flow calculated? Should it not be in %change from rest? As it stands right now, the absolute blood flow values and change in forearm blood flow are nearly the same absolute values; how can this be?

4. In Figure 5, extracellular ATP in isolated RBC from OA was not different versus saline and fasudil treatment. However, you state that fasudil improved isolated RBC ATP release.

5. In figure 2E, [ATP]_v in YA during hypoxia is significantly different versus saline and normoxia. However, fasudil did not improve [ATP]_v with hypoxia versus normoxia and fasudil. Why would fasudil improve the [ATP]_v with hypoxia in OA but not YA?

6. You showed that fasudil lowered blood pressure and improved forearm blood flow in OA. Fasudil and ROCK inhibition, has been shown to decrease blood pressure and circulating ANG-II, at least in an animal model of hypertension (Ocaranza et al. Hypertension, 2011). Given older adults typically have higher sympathetic activation, and thus likely elevated circulating angiotensin-II (ANG-II). Do you think that these fasudil-mediated changes in ANG-II may be occurring with your data and thus may be a potential mechanism for the hemodynamic improvements in OA?

MINOR CONCERNS

1. Fasudil is a vasodilator, what role do you think this fasudil-mediated vasodilation played in the ATP and hemodynamic responses herein? Is fasudil acting as the vasodilator and increasing ATP release or is fasudil increasing ATP release which then facilitates vasodilation?
2. Should VO₂ (ml/min) be normalized to relative VO₂ (ml/min/kg) for proper comparison between young and old?
3. This is very minor in my opinion, but consider consolidating the duplicated data presented in the tables and figures.
4. The introduction could be shortened and refined for brevity without losing the core information, and at the very least, more justification for the use of fasudil should be included.

END OF COMMENTS

Confidential Review

12-Dec-2021

EDITOR COMMENTS

Reviewing Editor:

The study supports a role of RBC rho-kinase in plasma ATP and blood flow, and the potential of Rho-kinase inhibition as an intervention that increases forearm blood flow during intense exercise in old individuals. Data are novel, nicely presented and discussed, and the study could have a significant impact on the field. The research group has a solid experience in the field. Both reviewers make precise comments and raise some issues (mainly related to data interpretation and analysis) which should be convincingly addressed by the authors.

Response: We thank the reviewing editor and referees for their positive comments regarding our study. We have addressed the concerns below, made revisions where appropriate, and believe the revised manuscript is strengthened as a result.

REFEREE COMMENTS

Referee #1:

GENERAL COMMENTS

This comprehensive study employed the rhythmic handgrip exercise model in conjunction with in vivo measurements of blood flow, blood oxygenation and plasma ATP in young and older individuals in normoxia, hypoxia and venous fasudil infusion and ex-vivo RBC ATP release in comparable conditions to investigate the impact of RBC rho-kinase inhibition in limb blood flow and circulating ATP. Interestingly, no absolute differences in forearm blood flow (FBF) or plasma [ATP] were observed between the older and young groups at rest in normoxia, at rest in hypoxia or during most of the subpeak rhythmic handgrip exercise bouts with either saline or fasudil infusion. The main positive findings were that systemic venous fasudil infusion induced 1) increases (above baseline) in FBF in hypoxia in both groups and 2) elevations in peak exercise hyperaemia in the older group (compared to baseline), but not in the young individuals. No absolute between group differences in FBF were observed at peak exercise with fasudil infusion. The study appears to have been well-conducted and the results with fasudil infusion and ex-vivo RBC ATP release are novel. That being said, the paper overemphasizes the positive results in the peak exercise condition and fails to provide the reader with information about the absolute differences in FBF as well as information about the conditions where FBF was not different between groups. This is important to put in perspective the magnitude of the phenomenon under investigation and thus provide insight into its therapeutic potential. There is also a need to clarify how arterial O₂ content and forearm VO₂ were estimated to convince the reader of the validity of the reported statistical differences in forearm VO₂, which are within the measurement error of the multiple variables used in this calculation. Lastly, it would be helpful to acknowledge in the experimental considerations and limitations section that further work is needed in older people undertaking other exercise modalities (including exercise capacity tests during larger muscle mass exercise) to ascertain the therapeutic potential of pharmacologically induced rho-kinase inhibition.

Response: Thank you for your positive comments as well as your suggestions for improvement. We have amended the manuscript accordingly.

SPECIFIC COMMENTS

TITLE - Specify that the rhythmic handgrip exercise model was employed. The effects of Rho-kinase inhibition during other exercise modalities remains unknown and thus the results cannot be generalised.

Response: Good point. Amended as suggested.

KEY POINTS - the background information tells the reader that a general impairment in skeletal muscle blood flow occurs with exposure to hypoxia and exercise, but this is not supported by the data of the present study, which only showed FBF differences at peak exercise in the saline infusion trial. The general statements in the first and last key points therefore need to be amended to specify the conditions where blood flow in older people is lower and thus its regulation might be 'impaired' (or altered) compared to young individuals.

Response: Impaired muscle blood flow and vascular conductance in older vs. young adults have been demonstrated by our laboratory and others during various exercise modalities (e.g., handgrip exercise, knee extensor exercise, cycling exercise) and various exercise intensities ranging from light to high intensity depending on the study (see review by Hearon Jr. & Dinunno, 2016). This is also the case when expressing the haemodynamic data as absolute values or a response from baseline (i.e., change/delta) to the exercise stimulus. Given this, we purposefully kept this statement general because it would be too difficult to specify the exact exercise conditions where this has been demonstrated particularly given the strict word limitations of the Key Points. Regarding the data in the present study, we agree with the Referee and made sure to emphasize “moderate intensity exercise” in the manuscript, as 15 and 25% MVC represent ~40 and ~70% WR_{max}, respectively (Richards *et al.*, 2014). Also, when the change in haemodynamics from baseline is different between groups of conditions, we use the term “response” to take into account individual differences in baseline blood flow and vascular conductance.

ABSTRACT - State whether the absolute exercise intensities were comparable between groups. As the authors know, this is critical to determine hemodynamic differences between groups, as the metabolic demand determined by the absolute work rate influences the magnitude of blood flow. Similarly, provide quantitative data of the magnitude of blood flow differences between groups to establish the physiological significance of the phenomenon.

Response: There were no differences in absolute exercise workloads and we added this briefly to the abstract. Given the amount of data collected in this study, it is not possible to add much quantitative data given the strict word limitations. Thus, we respectfully have chosen to keep the abstract as a comprehensive descriptive abstract.

ABSTRACT - Is the conclusion relating the improved circulating ATP and blood flow in older group based on any correlational analysis of FBF vs. plasma [ATP]?

Response: We have performed correlational analyses in several prior studies on this topic, and we have found it very difficult to interpret. As the reviewer is aware, venous plasma [ATP] is quite literally a “snapshot” of ATP concentrations at the level of a conduit vein draining the forearm circulation. We cannot measure this at the level of the microcirculation, which would be ideal. This, in combination with our understanding that the half-life of ATP in circulation is less than 0.5 seconds (Mortensen *et al.*, 2011), and is so rapid that infusion of ATP into the brachial artery (at a concentration that causes profound vasodilation) cannot be detected on the venous

side of the circulation (i.e., it is broken down in one pass of the circulation in this small tissue model in vivo; Kirby *et al.*, 2012), often results in poor correlations. Thus, we are cautious in emphasizing correlations with plasma ATP and recognize it is difficult to accurately measure (see page 20 of discussion). Mechanistic studies involving blockade of purinergic 2y (P2y) receptors or key signalling pathways downstream of P2y receptor stimulation will be needed to definitively address this issue.

INTRODUCTION - End of the introduction. Clarify why the rhythmic handgrip exercise is used to answer the question of the study.

Response: This was used based on prior observations in our lab for both haemodynamics (Kirby *et al.*, 2012; Richards *et al.*, 2014) and plasma ATP (Kirby *et al.*, 2012). We have also shown impaired haemodynamic and ATP responses to hypoxia in the forearm circulation (Kirby *et al.*, 2012). This has been added per your suggestion.

METHODS - Page 10. The absolute forearm vascular conductance values in this model are microliters/min/mmHg. Why do they need to be expressed per 100 mmHg?

Response: Our lab and others have used this approach to normalize the conductance per 100 mmHg which yields comparable values to blood flow (Tschakovsky *et al.*, 2002).

Page 10. The authors report the weights corresponding to the workloads used in the study in Table 1. Did you also measure the rate of contraction to obtain some index of power? The reported weight data do not seem to indicate that power was the same in both groups, as indicated by the statistical analysis.

Response: Given the similar workload in kg (P-values for the absolute workload comparison between young vs. older subjects at 5%, 15%, 25% MVC were 0.563, 0.463, and 0.499, respectively; individual subject data shown below) and identical load distance (0.035m) and duty cycle (20 contractions per minute) as monitored and performed with both audio and visual cues via metronome as done regularly in our lab, indices of work and power would be similar to the absolute load lifted.

Subject	Age	Sex	5% MVC (kg)	15% MVC (kg)	25% MVC (kg)
DM	Young	M	2.0	6.1	9.9
CW	Young	M	2.2	6.5	10.8
NM	Young	M	2.4	7.7	12.7
LL	Young	M	2.0	6.1	9.9
GM	Young	F	0.9	2.2	3.6
JR	Young	M	2.0	5.6	9.5
NH	Young	F	1.3	3.8	6.3
JT	Young	F	1.3	4.0	6.8
CK	Young	F	1.3	4.0	6.8
GR	Young	M	2.0	5.9	9.9
VJ	Young	F	1.3	4.3	7.2
BD	Young	F	0.9	2.4	4.3
SS	Older	F	1.1	3.1	5.2
BW	Older	M	2.2	6.5	10.8
JL	Older	F	1.1	2.9	4.9
DR	Older	M	1.8	5.2	8.6

DJ	Older	M	1.5	4.7	7.9
BeA	Older	M	1.8	5.2	8.6
BaA	Older	F	1.1	3.1	5.4
CR	Older	F	0.9	2.2	3.8
MW	Older	F	1.1	3.4	5.9
HH	Older	F	1.5	4.3	7.0
PS	Older	F	1.3	3.8	6.5
DF	Older	M	2.0	5.9	9.9
PT	Older	M	2.4	7.1	12.0

Page 14. The estimates of arterial O2 content are not clearly explained. Why should they be the same for all participants during exposure to normoxia (203 ml/l) and hypoxia (165 ml/l)? This has a bearing in the calculations of O2 delivery, O2 extraction and VO2, particularly when absolute values are relatively small compared to other exercise and environmental conditions. Please explain.

Response: Any estimate of arterial O2 content will have some degree of variability and error in the absence of a brachial artery catheter to directly measure arterial blood gases. For example, using pulse oximeter-derived SO2 data in the present study to estimate PO2 for calculating arterial O2 content provides PO2 estimates in normoxia that range from ~80mmHg to >145mmHg, which is far above direct measurements of arterial PO2 taken previously in our laboratory (Racine *et al.*, 2018). Thus, while we agree that all participants would not be expected to have the exact same arterial O2 content, we chose to utilize a consistent estimate of arterial O2 content across subjects based on direct brachial artery blood gas measurements from similar healthy young and older subjects who participated in similarly conducted prior studies within our laboratory (Richards *et al.*, 2014, 2017; no arterial O2 content differences with age) rather than more indirect estimates based on pulse oximetry.

RESULTS - Page 15. See comment above about workloads. Although 0.1, 0.5 and 0.7 kg 'lower' mean absolute workloads in the older group are not 'statistically' different from the young group, they can still affect blood flow values. See below the estimate of peak FBF per kg.

Response: We have added a statement to the results section (page 15, lines 15-18) to include this analysis. This impacts the age-group differences in FBF, but not FVC for the 25% MVC intensity. Importantly however, this normalization does not explain the fasudil-mediated improvements during exercise in older adults, nor the improved responses during hypoxia. While we understand this comment and the suggested normalization, this may reflect more math than physiology, and the proper test of this would require young and older adults performing an exercise at identical absolute workloads. In revision, we have been mindful and cautious in our data interpretation, and have highlighted the novelty of the data as it relates to the fasudil-mediated improvements in the older subjects, as this is the key finding and has never been demonstrated before.

	5% MVC		15% MVC		25% MVC	
	FBF/kg	FVC/kg	FBF/kg	FVC/kg	FBF/kg	FVC/kg
Young (saline)	44.2 ±11.3	49.7 ±15.9	39.2 ±5.4	44.3 ±6.9	41.8 ±6.9	45.7 ±9.6
Older (saline)	50.3 ±15.2	49.6 ±13.9	40.2 ±10.1	38.2 ±9.4	39.4 ±11.7	35.9 ±12.2
P-value	0.272	0.988	0.774	0.080	0.538	0.038

Page 16. Consider changing the increase in FBF and FVC was 'impaired' for 'was lower'.

Response: Amended accordingly.

Page 16. The magnitude of the phenomenon seems small when considering that forearm vascular conductance values were 9 vs. 6 microliters/min/mmHg.

Response: Yes, the haemodynamic changes to systemic hypoxia are modest and in proportion to degree of hypoxia/Hb desaturation. However, we have consistently shown similar impairments in older vs young adults (Kirby *et al.*, 2012; Richards *et al.*, 2017).

Page 16. This reviewer is not sure about the importance of the ATP effluent data when knowing that venous ATP infusion does not change limb flow. The level of plasma [ATP] is more relevant to explain blood flow changes in the present conditions. Blood flow is a confounding variable when relating changes in ATP effluent to changes in blood flow.

Response: While we agree that plasma [ATP] is important, we also believe that quantifying the circulating rate (ATP effluent) is important in that it reflects total ATP delivery (González-Alonso *et al.*, 2002), and we have discussed this at length previously (Kirby *et al.*, 2012). At present, it is not clear which expression of data is the most physiologically relevant, thus we present it both ways. This has been done for other variables measured in plasma when blood flow is elevated such as noradrenaline (Savard *et al.*, 1987; Proctor *et al.*, 1998) and tissue plasminogen activator (Pretorius *et al.*, 2003; Giannarelli *et al.*, 2009).

Page 17. Avoid using the word 'impaired' in the results section. This is an interpretation of the data, which is more suited for the discussion section. Here you could simply state whether data were higher, lower, or no different.

Response: Amended accordingly throughout the results.

Page 17. Consider normalising the data for the workload, as part of the differences in FBF, O₂ delivery and VO₂ can be accounted for by the lower absolute workload (0.7 kg at peak).

Response: We have responded to this above.

TABLES & FIGURES. There is duplication in the data presentation. To remedy this, figures could include not only the mean {plus minus} SD but also the individual data. The graphs with the delta data are not needed.

Response: We have removed some data from the Tables to address this issue. The response from baseline (delta) is an important means to express the data as it takes into account individual differences in baseline haemodynamics, thus we prefer to keep this form of data expression in the figures.

TABLES - Directly measured CvO₂ and [Hb] values could be reported below FBF and FVC data. The values of PvO₂ and PvCO₂ would also be helpful to know whether the experimental conditions of the ex-vivo experiments were comparable to those in the vivo experiment.

Response: PvO₂, PvCO₂, and a-vO₂ difference data have been added to Tables 2 and 3. [Hb] values have not been added to the tables as these data will not be comparable between *in vivo*

and *ex vivo* experiments due to the need to dilute RBC samples to a 20% haematocrit solution in order to properly measure extracellular ATP (Kirby *et al.*, 2012; Racine & Dinunno, 2019).

FIGURE 4. A graph illustrating the forearm a-vO₂ difference would be more insightful than O₂ extraction in discerning the factors accounting for the small differences in forearm VO₂ in some conditions.

Response: We have added a-vO₂ data to Tables 2 and 3, but have opted to keep O₂ extraction figures in the present manuscript to remain consistent with what our lab has published in the past (Crecelius *et al.*, 2011; Racine *et al.*, 2018).

DISCUSSION - Summary and other paragraphs interpreting the data. The conclusion paragraph reflects the findings, but does not provide an idea about the magnitude of the phenomenon under investigation, or make it clear to the reader that differences in FBF were only observed at peak exercise in the saline trial. This is crucial to be able to evaluate the potential implications of the study for older people undertaking other ecologically relevant exercise modalities. Statements of the magnitude of difference in FBF between older and young people during exercise and the variability in this and plasma [ATP] measurements are needed to address this concern.

Response: These are excellent points, and we have attempted to address these in revision throughout and in the experimental considerations section.

There is little discussion of the negative observations revealing similar between groups FBF at rest (normoxia and hypoxia) or during handgrip exercise at 5%MVC and 15%MVC with either saline or fasudil infusions. Between group FBF differences were only observed at 25% MVC (335 vs. 278 mL/min in older; $P < 0.05$). This effect during the saline condition could be in part confounded by differences in workload (8.1 vs 7.4 kg; FBF per kg = 41 vs. 38 ml/min/kg in young vs. older group). Please clarify whether the normalised data would still be statistically different.

Response: Similar FBF and FVC at rest is an expected finding as our lab and others have shown this repeatedly. While the absolute FBF/FVC are not different between groups during hypoxia, the response (Δ) is different and important when considering individual differences in baseline haemodynamics. We addressed the normalisation data above. Importantly, normalising the data does not explain the fasudil-mediated improvements in FBF/FVC responses during hypoxia and moderate intensity exercise in older adults.

Referee #2:

The manuscript by Racine et al describes potential mechanisms to ameliorate age-induced impairments in skeletal muscle vascular function and hemodynamic responses through improvements in circulating ATP during both hypoxia and hand grip exercise. The authors demonstrate that decreased ATP release may be involved in the increased risk of CVD and the decline in functional capacity with primary aging. Further, the use of Fasudil may alleviate age-associated CVD risk and/or the declines in functional capacity. The experimental paradigm is well designed, and the manuscript is well written. However, there are some concerns with the data and analysis and aspects of the discussion that must be addressed. Below are major and minor concerns.

Response: Thank you for the positive comments and enthusiasm for our study. We have addressed your concerns below and in the revised manuscript.

MAJOR CONCERNS

1. In the key points section, you state "...fasudil, improved blood flow and circulating ATP responses during hypoxia and hand grip exercise compared to adults...". However, in Figure 2, this does not appear to be the case during hypoxia. Unless I am missing something, there is no comparison made between older adults (OA) and young adults (YA).

Response: We did in fact compare the FBF/FVC responses (Δ FBF/FVC) between young and older (e.g., Figure 2C and D) and also with saline or fasudil. This was also done for circulating ATP (ATP effluent, Figure 2F).

2. In figure 2E, OA [ATP]v with fasudil and hypoxia is significantly different versus fasudil and normoxia, however, in table 3, they are not significantly different. With hypoxia and fasudil versus hypoxia and saline in OA, [ATP]v is not different. Further, in figure 2F, hypoxia and fasudil did not increase ATP effluent versus saline and hypoxia in OA. Please explain these discrepancies and lack of significant difference with Fasudil versus saline, as this is clearly a primary outcome.

Response: In Figure 2E, [ATP]v tended to be greater in hypoxia with fasudil vs saline in the older adults ($P=0.073$). Table 3 shows data from the exercise trials, so we are unclear about this comment as it relates to hypoxia. In Table 2, the data is shown as not significant, but again the P-value was 0.07. Per Referee #1's suggestion, and that it is difficult working back and forth through the figures and tables, we have removed absolute FBF, FVC and [ATP]v from the tables as these are shown in Figures 2 and 3. In Figure 2F, ATP effluent was slightly lower at baseline with fasudil in the older subjects. ATP effluent was significantly increased during hypoxia in older adults within condition ($P<0.0001$), but this value did not achieve statistical significance compared with that observed at baseline (normoxia) with saline ($P=0.251$).

3. Figure 3A-D: How was delta blood flow calculated? Should it not be in %change from rest? As it stands right now, the absolute blood flow values and change in forearm blood flow are nearly the same absolute values; how can this be?

Response: Delta blood flow was calculated as either hypoxia blood flow or exercise blood flow (for each intensity) minus resting blood flow within condition. There are many ways to express the data. Given that there were no age group differences in resting forearm blood flow (or vascular conductance), we opted to present the absolute change in FBF to account for individual differences. Given that resting FBF is relatively low (~20 ml/min), the values for the change in blood flow are quite similar to absolute blood flow since the increase in FBF is quite large during exercise.

4. In Figure 5, extracellular ATP in isolated RBC from OA was not different versus saline and fasudil treatment. However, you state that fasudil improved isolated RBC ATP release.

Response: This is based on the comparison within condition. Extracellular ATP was not significantly increased above baseline in response to hypoxia in the saline condition, but was in the fasudil condition (Figure 5A). We have presented this in a different form in Figure 5B and have described this in the respective paragraph in the results (page 16).

5. In figure 2E, [ATP]_v in YA during hypoxia is significantly different versus saline and normoxia. However, fasudil did not improve [ATP]_v with hypoxia versus normoxia and fasudil. Why would fasudil improve the [ATP]_v with hypoxia in OA but not YA?

Response: This is the primary hypothesis of the study. We have shown previously that red blood cells from older adults are less deformable and release less ATP during hypoxia than young adults, and treatment in vitro with a rho kinase inhibitor improves deformability and ATP release in older adults only. Thus, we were attempting to translate these findings to an *in vivo* model. The present data, to some extent, support this hypothesis.

6. You showed that fasudil lowered blood pressure and improved forearm blood flow in OA. Fasudil and ROCK inhibition, has been shown to decrease blood pressure and circulating ANG-II, at least in an animal model of hypertension (Ocaranza et al. Hypertension, 2011). Given older adults typically have higher sympathetic activation, and thus likely elevated circulating angiotensin-II (ANG-II). Do you think that these fasudil-mediated changes in ANG-II may be occurring with your data and thus may be a potential mechanism for the hemodynamic improvements in OA?

Response: Interesting point. This is possible, however we believe this is unlikely as our previous studies demonstrate that the age-related impairments in blood flow responses to hypoxia or graded exercise are not due to elevated sympathetic nervous system activity or greater sympathetically-mediated vasoconstriction (Richards et al., 2014, 2017). We have not performed studies with ANG-II blockade, but we do not believe this is likely.

MINOR CONCERNS

1. Fasudil is a vasodilator, what role do you think this fasudil-mediated vasodilation played in the ATP and hemodynamic responses herein? Is fasudil acting as the vasodilator and increasing ATP release or is fasudil increasing ATP release which then facilitates vasodilation?

Response: Great question, and from the present study design, it is not possible to definitively address this. We don't believe that the possible direct vasodilator effect of fasudil explains the findings for a few reasons. First, baseline FBF and FVC were not significantly different with fasudil vs. saline, and the responses were only different during the hypoxic or exercise stimuli. As for venous plasma [ATP], previous studies from our laboratory and others have shown that vasodilation *per se* does not increase plasma [ATP] (Mortensen et al., 2011; Kirby et al., 2012), and may actually decrease it due to potential dilution effects of elevated blood flow.

2. Should VO₂ (ml/min) be normalized to relative VO₂ (ml/min/kg) for proper comparison between young and old?

Response: There were no differences in forearm fat-free mass (Table 1), thus we elect to present these as absolute values.

3. This is very minor in my opinion, but consider consolidating the duplicated data presented in the tables and figures.

Response: Thank you for this suggestion. We have amended the revised manuscript accordingly.

4. The introduction could be shortened and refined for brevity without losing the core

information, and at the very least, more justification for the use of fasudil should be included.

Response: We have looked at the introduction again and believe that the length is appropriate given the integrative topic at hand. We have attempted to clarify and justify the use of fasudil in revision. This is based on our recent observations that treatment of isolated red blood cells from older adults with a Rho-kinase inhibitor in vitro improves ATP release during hypoxia.

References

- Crecelius AR, Kirby BS, Voyles WF & Dinunno FA (2011). Augmented skeletal muscle hyperaemia during hypoxic exercise in humans is blunted by combined inhibition of nitric oxide and vasodilating prostaglandins. *J Physiol* **589**, 3671–3683.
- Giannarelli C, Viridis A, De Negri F, Magagna A, Duranti E, Salvetti A & Taddei S (2009). Effect of sulfaphenazole on tissue plasminogen activator release in normotensive subjects and hypertensive patients. *Circulation* **119**, 1625–1633.
- González-Alonso J, Olsen DB & Saltin B (2002). Erythrocyte and the regulation of human skeletal muscle blood flow and oxygen delivery: role of circulating ATP. *Circ Res* **91**, 1046–1055.
- Hearon Jr. C & Dinunno F (2016). Regulation of skeletal muscle blood flow during exercise in ageing humans. *J Physiol* **594**, 2261–2273.
- Kirby BS, Crecelius AR, Voyles WF & Dinunno FA (2012). Impaired skeletal muscle blood flow control with advancing age in humans: attenuated ATP release and local vasodilation during erythrocyte deoxygenation. *Circ Res* **111**, 220–230.
- Mortensen SP, Thaning P, Nyberg M, Saltin B & Hellsten Y (2011). Local release of ATP into the arterial inflow and venous drainage of human skeletal muscle: insight from ATP determination with the intravascular microdialysis technique. *J Physiol* **589**, 1847–1857.
- Pretorius M, Rosenbaum D, Vaughan DE & Brown NJ (2003). Angiotensin-converting enzyme inhibition increases human vascular tissue-type plasminogen activator release through endogenous bradykinin. *Circulation* **107**, 579–585.
- Proctor DN, Shen PH, Dietz NM, Eickhoff TJ, Lawler LA, Ebersold EJ, Loeffler DL & Joyner MJ (1998). Reduced leg blood flow during dynamic exercise in older endurance-trained men. *J Appl Physiol* **85**, 68–75.
- Racine ML, Crecelius AR, Luckasen GJ, Larson DG & Dinunno FA (2018). Inhibition of Na⁺/K⁺-ATPase and KIR channels abolishes hypoxic hyperaemia in resting but not contracting skeletal muscle of humans. *J Physiol* **596**, 3371–3389.
- Racine ML & Dinunno FA (2019). Reduced deformability contributes to impaired deoxygenation-induced ATP release from red blood cells of older adult humans. *J Physiol* **597**, 4503–4519.
- Richards JC, Crecelius AR, Larson DG, Luckasen GJ & Dinunno FA (2017). Impaired peripheral vasodilation during graded systemic hypoxia in healthy older adults: role of the sympathoadrenal system. *Am J Physiol - Hear Circ Physiol* **312**, H832–H841.
- Richards JC, Luckasen GJ, Larson DG & Dinunno FA (2014). Role of α -adrenergic vasoconstriction in regulating skeletal muscle blood flow and vascular conductance during forearm exercise in ageing humans. *J Physiol* **21**, 4775–4788.
- Savard G, Strange S, Kiens B, Richter EA, Christensen NJ & Saltin B (1987). Noradrenaline spillover during exercise in active versus resting skeletal muscle in man. *Acta Physiol Scand* **131**, 507–515.
- Tschakovsky ME, Sujirattanawimol K, Ruble SB, Valic Z & Joyner MJ (2002). Is sympathetic neural vasoconstriction blunted in the vascular bed of exercising human muscle? *J Physiol* **541**, 623–635.

Dear Dr Dinunno,

Re: JP-RP-2022-282730R1 "Rho-kinase inhibition improves haemodynamic responses & circulating ATP during hypoxia & moderate intensity handgrip exercise in healthy older adults" by Matthew L. Racine, Janée D. Terwoord, Nathaniel B. Ketelhut, Nate P. Bachman, Jennifer C. Richards, Gary J Luckasen, and Frank A. Dinunno

Thank you for submitting your revised Research Article to The Journal of Physiology. It has been assessed by the original Reviewing Editor and Referees and has been well received. Some final revisions have been requested.

The reports are copied at the end of this email. Please address all of the points and incorporate all requested revisions, or explain in your Response to Referees why a change has not been made.

NEW POLICY: In order to improve the transparency of its peer review process The Journal of Physiology publishes online as supporting information the peer review history of all articles accepted for publication. Readers will have access to decision letters, including all Editors' comments and referee reports, for each version of the manuscript and any author responses to peer review comments. Referees can decide whether or not they wish to be named on the peer review history document.

Authors are asked to use The Journal's premium BioRender (<https://biorender.com/>) account to create/redraw their Abstract Figures. Information on how to access The Journal's premium BioRender account is here: <https://physoc.onlinelibrary.wiley.com/journal/14697793/biorender-access> and authors are expected to use this service. This will enable Authors to download high-resolution versions of their figures. The link provided should only be used for the purposes of this submission. Authors will be charged for figures created on this premium BioRender account if they are not related to this manuscript submission.

I hope you will find the comments helpful and have no difficulty returning your revisions within one week.

Your revised manuscript should be submitted online using the links in Author Tasks Link Not Available.

Any image files uploaded with the previous version are retained on the system. Please ensure you replace or remove all files that have been revised.

REVISION CHECKLIST:

- Article file, including any tables and figure legends, must be in an editable format (eg Word)
- Abstract figure file (see above)
- Statistical Summary Document
- Upload each figure as a separate high quality file
- Upload a full Response to Referees, including a response to any Senior and Reviewing Editor Comments;
- Upload a copy of the manuscript with the changes highlighted.

- A potential 'Cover Art' file for consideration as the Issue's cover image;
- Appropriate Supporting Information (Video, audio or data set https://jp.msubmit.net/cgi-bin/main.plex?form_type=display_requirements#supp).

To create your 'Response to Referees' copy all the reports, including any comments from the Senior and Reviewing Editors, into a Word, or similar, file and respond to each point in colour or CAPITALS and upload this when you submit your revision.

I look forward to receiving your revised submission.

If you have any queries please reply to this email and staff will be happy to assist.

Yours sincerely,

Michael C. Hogan
Senior Editor
The Journal of Physiology
<https://jp.msubmit.net>
<http://jp.physoc.org>
The Physiological Society
Hodgkin Huxley House
30 Farringdon Lane
London, EC1R 3AW
UK
<http://www.physoc.org>
<http://journals.physoc.org>

-

EDITOR COMMENTS

Reviewing Editor:

The study provides novel evidence that Rho-inhibition is associated with increases in limb blood flow and circulating ATP during hypoxia and moderate intensity handgrip exercise in healthy older individuals. The topic is of interest and both referees expressed positive comments. Whereas Reviewer 2 is satisfied with the revision, Reviewer 1 still has one pertinent comment which the authors should take into consideration. The Reviewer suggests to estimate CaO₂ values from data of the present study (and to recognize the limitation associated with the estimate), without taking CaO₂ values from a previous study.

REFEREE COMMENTS

Referee #1:

This reviewer appreciates the thorough responses to the comments and the corresponding amendments in the manuscript. The clarity of the manuscript has been improved as the result. There is, however, a remaining concern that needs further consideration, namely the use of CaO₂ mean data from a previous study to calculate O₂ delivery and forearm VO₂ in the present study. A stronger approach is to use the present directly measured individual venous [Hb] and SpO₂ values to estimate the CaO₂ values for each participant and condition. The corresponding PaO₂ can be estimated from a standard O₂ dissociation curve. The latter will not have a major impact in CaO₂ in comparison to differences in [Hb] and O₂Hb among participants and conditions. The fact that this is an estimate should be mentioned in the experimental considerations section of the discussion in particular in relation to the reported findings based on estimates of forearm VO₂. Lastly, it is recommended to report [Hb]_v in tables 2 and 3, as done for the isolated RBC experiment in Table 4.

Referee #2:

Overall, this is an impactful study and is of high-impact to the field. Well done.

END OF COMMENTS

1st Confidential Review

31-Mar-2022

EDITOR COMMENTS

Reviewing Editor:

The study provides novel evidence that Rho-inhibition is associated with increases in limb blood flow and circulating ATP during hypoxia and moderate intensity handgrip exercise in healthy older individuals. The topic is of interest and both referees expressed positive comments. Whereas Reviewer 2 is satisfied with the revision, Reviewer 1 still has one pertinent comment which the authors should take into consideration. The Reviewer suggests to estimate CaO₂ values from data of the present study (and to recognize the limitation associated with the estimate), without taking CaO₂ values from a previous study.

REFEREE COMMENTS

Referee #1:

This reviewer appreciates the thorough responses to the comments and the corresponding amendments in the manuscript. The clarity of the manuscript has been improved as the result. There is, however, a remaining concern that needs further consideration, namely the use of CaO₂ mean data from a previous study to calculate O₂ delivery and forearm VO₂ in the present study. A stronger approach is to use the present directly measured individual venous [Hb] and SpO₂ values to estimate the CaO₂ values for each participant and condition. The corresponding PaO₂ can be estimated from a standard O₂ dissociation curve. The latter will not have a major impact in CaO₂ in comparison to differences in [Hb] and O₂Hb among participants and conditions. The fact that this is an estimate should be mentioned in the experimental considerations section of the discussion in particular in relation to the reported findings based on estimates of forearm VO₂. Lastly, it is recommended to report [Hb]_v in tables 2 and 3, as done for the isolated RBC experiment in Table 4.

Response: We have calculated CaO₂ in this manner per your request, and have added [Hb]_v to Tables 2 and 3. We have also included the following statement on page 18 regarding blood gasses: "Given that we estimated CaO₂ in the present study and that blood gasses were not obtained in all subjects, caution should be taken when interpreting the findings related to forearm $\dot{V}O_2$." All changes to the revised manuscript are in red font. We hope you find our manuscript now acceptable for publication, and we appreciate your time, efforts, and critical yet positive feedback on our work.

Referee #2:

Overall, this is an impactful study and is of high-impact to the field. Well done.

Response: Thank you!

Dear Dr Dinunno,

Re: JP-RP-2022-282730R2 "Rho-kinase inhibition improves haemodynamic responses & circulating ATP during hypoxia & moderate intensity handgrip exercise in healthy older adults" by Matthew L. Racine, Janée D. Terwoord, Nathaniel B. Ketelhut, Nate P. Bachman, Jennifer C. Richards, Gary J Luckasen, and Frank A. Dinunno

Thank you for submitting your revised Research Article to The Journal of Physiology. It has been assessed by the original Reviewing Editor and Referees and has been well received. Some final revisions have been requested.

The reports are copied at the end of this email. Please address all of the points and incorporate all requested revisions, or explain in your Response to Referees why a change has not been made.

NEW POLICY: In order to improve the transparency of its peer review process The Journal of Physiology publishes online as supporting information the peer review history of all articles accepted for publication. Readers will have access to decision letters, including all Editors' comments and referee reports, for each version of the manuscript and any author responses to peer review comments. Referees can decide whether or not they wish to be named on the peer review history document.

Authors are asked to use The Journal's premium BioRender (<https://biorender.com/>) account to create/redraw their Abstract Figures. Information on how to access The Journal's premium BioRender account is here: <https://physoc.onlinelibrary.wiley.com/journal/14697793/biorender-access> and authors are expected to use this service. This will enable Authors to download high-resolution versions of their figures. The link provided should only be used for the purposes of this submission. Authors will be charged for figures created on this premium BioRender account if they are not related to this manuscript submission.

I hope you will find the comments helpful and have no difficulty returning your revisions within one week.

Your revised manuscript should be submitted online using the links in Author Tasks Link Not Available.

Any image files uploaded with the previous version are retained on the system. Please ensure you replace or remove all files that have been revised.

REVISION CHECKLIST:

- Article file, including any tables and figure legends, must be in an editable format (eg Word)
- Abstract figure file (see above)
- Statistical Summary Document
- Upload each figure as a separate high quality file
- Upload a full Response to Referees, including a response to any Senior and Reviewing Editor Comments;
- Upload a copy of the manuscript with the changes highlighted.

- A potential 'Cover Art' file for consideration as the Issue's cover image;
- Appropriate Supporting Information (Video, audio or data set https://jp.msubmit.net/cgi-bin/main.plex?form_type=display_requirements#supp).

To create your 'Response to Referees' copy all the reports, including any comments from the Senior and Reviewing Editors, into a Word, or similar, file and respond to each point in colour or CAPITALS and upload this when you submit your revision.

I look forward to receiving your revised submission.

If you have any queries please reply to this email and staff will be happy to assist.

Yours sincerely,

Michael C. Hogan
Senior Editor
The Journal of Physiology
<https://jp.msubmit.net>
<http://jp.physoc.org>
The Physiological Society
Hodgkin Huxley House
30 Farringdon Lane
London, EC1R 3AW
UK
<http://www.physoc.org>
<http://journals.physoc.org>

EDITOR COMMENTS

Reviewing Editor:

Whereas Reviewer 2 is happy with the revisions, Reviewer 1 still asks for a clarification.

REFeree COMMENTS

Referee #1:

The reviewer appreciates the additional work. It remains unclear, however, whether the authors have actually updated all the calculated variables that depend on the estimated CaO_2 . For instance, the $a\text{-vO}_2$ difference data reported in the tables are identical in the revised version and the first submission. Please clarify whether the recalculated forearm $a\text{-vO}_2$ diff, O_2 delivery, O_2 extraction and VO_2 data have been included in the revised manuscript.

Referee #2:

All my concerns have been addressed.

END OF COMMENTS

2nd Confidential Review

20-Apr-2022

EDITOR COMMENTS

Reviewing Editor:

Whereas Reviewer 2 is happy with the revisions, Reviewer 1 still asks for a clarification.

REFEREE COMMENTS

Referee #1:

The reviewer appreciates the additional work. It remains unclear, however, whether the authors have actually updated all the calculated variables that depend on the estimated CaO₂. For instance, the a-vO₂ difference data reported in the tables are identical in the revised version and the first submission. Please clarify whether the recalculated forearm a-vO₂diff, O₂ delivery, O₂ extraction and VO₂ data have been included in the revised manuscript.

Response: Our apologies for the mistaken omission of the a-vO₂ difference data from Tables 2 and 3. The recalculated O₂ delivery, O₂ extraction and VO₂ data were included in the previous resubmission in the results section (page 16) and in Figure 4 (page 38), but you are correct that the recalculated a-vO₂ difference data were not updated accordingly in Tables 2 and 3. These data have now been added to the tables and are highlighted in red font. Please accept our sincere apologies and thank you again for your time, efforts, and feedback on our work.

Referee #2:

All my concerns have been addressed.

Response: Thank you.

Dear Dr Dinenzo,

Re: JP-RP-2022-282730R3 "Rho-kinase inhibition improves haemodynamic responses & circulating ATP during hypoxia & moderate intensity handgrip exercise in healthy older adults" by Matthew L. Racine, Janée D. Terwoord, Nathaniel B. Ketelhut, Nate P. Bachman, Jennifer C. Richards, Gary J Luckasen, and Frank A. Dinenzo

I am pleased to tell you that your paper has been accepted for publication in The Journal of Physiology, subject to any modifications to the text and/or satisfactory clarification of the Methods section that may be required by the Journal Office to conform to House rules.

NEW POLICY: In order to improve the transparency of its peer review process The Journal of Physiology publishes online as supporting information the peer review history of all articles accepted for publication. Readers will have access to decision letters, including all Editors' comments and referee reports, for each version of the manuscript and any author responses to peer review comments. Referees can decide whether or not they wish to be named on the peer review history document.

The last Word version of the paper submitted will be used by the Production Editors to prepare your proof. When this is ready you will receive an email containing a link to Wiley's Online Proofing System. The proof should be checked and corrected as quickly as possible.

Authors should note that it is too late at this point to offer corrections prior to proofing. Major corrections at proof stage, such as changes to figures, will be referred to the Reviewing Editor for approval before they can be incorporated. Only minor changes, such as to style and consistency, should be made a proof stage. Changes that need to be made after proof stage will usually require a formal correction notice.

All queries at proof stage should be sent to TJP@wiley.com

The accepted version of the manuscript will be published online, prior to copy editing, in the Accepted Articles section.

Are you on Twitter? Once your paper is online, why not share your achievement with your followers. Please tag The Journal (@jphysiol) in any tweets and we will share your accepted paper with our 22,000+ followers!

Yours sincerely,

Michael C. Hogan
Senior Editor
The Journal of Physiology
<https://jp.msubmit.net>
<http://jp.physoc.org>
The Physiological Society
Hodgkin Huxley House
30 Farringdon Lane
London, EC1R 3AW
UK
<http://www.physoc.org>
<http://journals.physoc.org>

P.S. - You can help your research get the attention it deserves! Check out Wiley's free Promotion Guide for best-practice recommendations for promoting your work at www.wileyauthors.com/eeo/guide. And learn more about Wiley Editing Services which offers professional video, design, and writing services to create shareable video abstracts, infographics, conference posters, lay summaries, and research news stories for your research at www.wileyauthors.com/eeo/promotion.

* IMPORTANT NOTICE ABOUT OPEN ACCESS *

Information about Open Access policies can be found here <https://physoc.onlinelibrary.wiley.com/hub/access-policies>

To assist authors whose funding agencies mandate public access to published research findings sooner than 12 months after publication The Journal of Physiology allows authors to pay an open access (OA) fee to have their papers made freely available immediately on publication.

You will receive an email from Wiley with details on how to register or log-in to Wiley Authors Services where you will be able to place an OnlineOpen order.

You can check if your funder or institution has a Wiley Open Access Account here <https://authorservices.wiley.com/author-resources/Journal-Authors/licensing-and-open-access/open-access/author-compliance-tool.html>

Your article will be made Open Access upon publication, or as soon as payment is received.

If you wish to put your paper on an OA website such as PMC or UKPMC or your institutional repository within 12 months of publication you must pay the open access fee, which covers the cost of publication.

OnlineOpen articles are deposited in PubMed Central (PMC) and PMC mirror sites. Authors of OnlineOpen articles are permitted to post the final, published PDF of their article on a website, institutional repository, or other free public server, immediately on publication.

Note to NIH-funded authors: The Journal of Physiology is published on PMC 12 months after publication, NIH-funded authors DO NOT NEED to pay to publish and DO NOT NEED to post their accepted papers on PMC.

EDITOR COMMENTS

The remaining reviewer is satisfied. Congratulations for an excellent study.

REFEREE COMMENTS

Referee #1:

The authors have adequately addressed my last comment.